# Macrocycle-based PROTACs selectively degrade cyclophilin A and inhibit HIV-1 and HCV

Lydia S. Newton [1,8], Clara Gathmann[2,8], Sophie Ridewood[1], Robert J. Smith [2], Andre J. Wijaya [3], Thomas W. Hornsby[2], Kate L. Morling [1,2], Dara Annett[1], Riccardo Zenezini Chiozzi [4], Ann-Kathrin Reuschl [1], Morten L. Govasli[1,5], Ying Ying Tan[1], Lucy G. Thorne [1,6], Clare Jolly [1], Konstantinos Thalassinos [4,7], Alessio Ciulli [3], Greg J. Towers [1] ✉ & David L. Selwood [2] ✉

Targeting host proteins that are crucial for viral replication offers a promising antiviral strategy. We have designed and characterised antiviral PROteolysis TArgeting Chimeras (PROTACs) targeting the human protein cyclophilin A (CypA), a host cofactor for unrelated viruses including human immunodeficiency virus (HIV) and hepatitis C virus (HCV). The PROTAC warheads are based on fully synthetic macrocycles derived from sanglifehrin A, which are structurally different from the classical Cyp inhibitor, cyclosporine A. Our Cyp-PROTACs decrease CypA levels in cell lines and primary human cells and have high specificity for CypA confirmed by proteomics experiments. Critically, CypA degradation facilitates improved antiviral activity against HIV-1 in primary human CD4+ T cells compared to the non-PROTAC parental inhibitor, at limiting inhibitor concentrations. Similarly, we observe antiviral activity against HCV replicon in a hepatoma cell line. We propose that CypA-targeting PROTACs inhibit viral replication potently and anticipate reduced evolution of viral resistance and broad efficacy against unrelated viruses. Furthermore, they provide powerful tools for probing cyclophilin biology.

Antiviral drugs typically target viral proteins, but disadvantages include the rapid evolution of resistance mutations and narrow inhibitor specificity against closely related viruses. Targeting host proteins essential for viral replication, also known as cofactors, is an alternative and under-explored approach. Unrelated viruses often depend on the same host proteins for replication making cofactors potential targets for broad-spectrum antivirals. Cofactor targeting is expected to have a higher barrier to resistance because the inhibitor does not contact the viral proteome directly[1]. While many viral cofactors have been described, few have been clinically exploited[2]. Hesitancy derives from toxicity risks associated with host targeting and theoretical risks of driving resistance in unrelated subclinical infections. However, the recent COVID-19 and influenza H1N1 pandemics have underlined the need for ready-to-go, broad-spectrum antivirals active against novel pandemics[3,4].

The development of targeted protein degradation brings exciting advantages over competitive inhibitors. PROteolysis TArgeting Chimeras (PROTACs) drive ubiquitin-dependent degradation of selected

[1]Division of Infection and Immunity, University College London, London, UK. [2]Wolfson Institute for Biomedical Research, University College London, London, UK. [3]Centre for Targeted Protein Degradation, School of Life Sciences, University of Dundee, Dundee, UK. [4]University College London Mass Spectrometry Science Technology Platform, Division of Biosciences, University College London, London, UK. [5]Department of Biomedicine, Centre for Cancer Biomarkers, University of Bergen, Bergen, Norway. [6]Department of Infectious Diseases, Imperial College London, London, UK. [7]Institute of Structural and Molecular Biology, Division of Biosciences, University College London, London, UK. [8]These authors contributed equally: Lydia S. Newton, Clara Gathmann. ✉e-mail: g.towers@ucl.ac.uk; d.selwood@ucl.ac.uk

cellular proteins by forming complexes between ubiquitin ligases and target proteins. This catalytic mechanism typically results in prolonged inhibition and increased therapeutic potency compared to stoichiometric competitive inhibitors[5,6]. Antiviral PROTAC development has also gained interest for its potential to reduce sensitivity to resistance[7–10]. Here, we sought to combine PROTAC-mediated degradation with host cofactor targeting as a proof-of-concept antiviral strategy.

Cyclophilins (Cyps) are attractive targets for broad-spectrum antivirals[11,12] because divergent viruses depend on them as cofactors, including human immunodeficiency virus (HIV)[13] and hepatitis C virus (HCV)[14]. There is also evidence for the involvement of CypA in dengue virus (DENV)[15], coronaviruses including SARS-CoV[16], enteroviruses[17] and hepatitis B virus (HBV)[18]. Cyps are peptidyl-prolyl isomerases that induce conformational changes in target proteins and act as chaperones, a role that impacts unrelated biological processes[19]. Multiple Cyp isoforms exist with cytosolic CypA being the most abundant and best understood. CypA is likely to have different roles for different viruses, but in the best-studied examples of HIV and HCV, CypA has roles in protecting infection from host innate immune responses[20,21]. For example, an exposed proline-rich loop on the HIV-1 capsid (CA) surface binds to CypA, regulating capsid dynamics[22] and shielding subsequent protein interactions including the antiviral protein TRIM5α[23–25]. When CypA-CA interactions are inhibited, for example with the CypA inhibitor cyclosporine A (CsA)[26], human TRIM5α targets the HIV-1 capsid and inhibits infectivity by forming a cage-like structure around the viral core. Caging drives TRIM5α trimerisation, autoubiquitination and innate immune sensing, leading to the induction of an antiviral state[23,25,27]. In the case of HCV, CypA interacts with the proline-rich region of domain 2 of NS5A, a non-structural HCV protein that mediates viral replication and assembly[28,29]. This interaction enhances RNA binding and supports the formation of double-membrane vesicles (DMVs), thus contributing to viral cloaking. This is evidenced by Cyp inhibitor treatment resulting in decreased size and number of DMVs[21,30,31] and causing enhanced expression of interferon stimulated genes (ISGs) in HCV-infected Huh7 cells[21]. Importantly, Colgan et al. have suggested that CypA is not essential in mammals, evidenced by CypA knockout (Ppia⁻/⁻) mice having a normal lifespan[32], supporting its suitability as an antiviral PROTAC target.

The classical cyclophilin inhibitor CsA is not suitable as an antiviral due to potent immunosuppressive activity mediated via inhibition of the T cell phosphatase calcineurin[33]. However, well-characterised, non-immunosuppressive variants of CsA have been developed, including NIM-811, alisporivir (DEB025), SCY-635 and CRV431 (CPI-431-32/rencofilstat), all with activity against HIV-1 and HCV replication[34,35]. Alisporivir, the most promising, only had limited clinical efficacy against HIV-1 in HIV-1/HCV co-infected patients[36] and toxicity issues halted HCV antiviral trials in 2012[37]. Although these studies establish cyclophilins as candidate antiviral drug targets, Cyp inhibitors have yet to be brought to market, with off-target effects seen as an important problem[38,39]. In addition, the lack of clinical efficacy compared to highly potent virus-targeting inhibitors[40] has reduced enthusiasm for targeting HIV-1 and HCV with Cyp inhibitors. Recent clinical studies have nevertheless investigated CRV431 against HBV (NCT03596697) and alisporivir against SARS-CoV-2 (NCT04608214).

While CsA-based PROTACs may address potency challenges, CsA synthesis is complex, and semi-synthesis is limited in scope[41,42], impairing medicinal chemistry optimisation. However, additional cyclophilin inhibitors have also been developed, for example, based on the natural product sanglifehrin A[43] (SfA, Fig. 1a), a low nanomolar Cyp inhibitor that also exerts immunosuppression but via a calcineurin-independent pathway[44]. SfA and its derivatives show comparable potency to CsA against HCV replicons in Huh7 cells[45] and improved potency in vitro against HIV-1[46]. SfA comprises a tripeptidic macrocyclic core attached to a polyketidic chain containing a spirocyclic scaffold

(Fig. 1a). Structural studies of the SfA-CypA complex suggested that only the macrolide is required for nanomolar Cyp binding, and removal of the polyketidic chain abrogates immunosuppressive activity[47]. Based on this finding, truncated semi-synthetic, non-immunosuppressive sangamides have been developed (NVP018, Supplementary Fig. 1S2)[48] with picomolar Cyp inhibition and antiviral activity against HBV, HIV-1 and HCV[49,50]. Further pioneering, iterative simplification of the SfA macrocycle guided by biological and crystallographic studies led to fully synthetically accessible nanomolar Cyp inhibitors (Supplementary Fig. 1S3 and S4). These molecules have optimised pharmacokinetic properties and are effective against HCV replicons[51,52], while anti-HIV-1 activity is only preliminary[23]. Importantly, the sanglifehrin class of inhibitors has reduced off-target inhibition of transporter proteins compared to CsA-based inhibitors[49]. Therefore, these fully synthetic molecules hold promise as a new class of Cyp-targeting antivirals.

Here, we report the development and characterisation of two fully synthetic cyclophilin PROTACs (Cyp-PROTACs), CG167 and RJS308, that degrade CypA to exert antiviral activity. They are based on a new SfA-inspired macrocyclic Cyp inhibitor, TWH106, which is structurally distinct from CsA (Fig. 1a). We demonstrate that CypA degradation is driven by a PROTAC mechanism and evidence CypA selectivity in different cell lines, primary human monocyte-derived macrophages and CD4+ T cells. These Cyp-PROTACs have improved antiviral activity against HIV-1 and HCV compared to the non-PROTAC parental inhibitor TWH106. Our molecules provide a proof of concept for PROTACs targeting CypA to inhibit infection and will be useful tools to probe isoform-specific cyclophilin biology.

## Results
### Design of PROTACs based on the TWH106 macrocycle with high affinity for cyclophilins

Previously to our work, simplified SfA-related macrocycles (Supplementary Fig. 1S3 and S4) were designed with an aryl or heteroaryl group substituting the conjugated olefinic unit in SfA, enabling a π-stacking interaction with CypA residue R55[51,52]. In addition, the SfA piperazic acid (pip¹) – m-tyrosine (mTyr²) – valine (Val³) sequence (Fig. 1a) was truncated to pip¹-Ala²-Val³ as the mTyr was not essential for binding[52]. Using this scaffold as a starting point, we reasoned that the sub-pocket near CypA residue K82 and E81 (Fig. 1a) could be filled with an aromatic residue at position 3, offering a synthetic handle for potency improvement and selectivity tuning. Importantly, this sub-pocket is the only region of the Cyp active sites that varies across cyclophilin isoforms[53] (Supplementary Fig. 8a).

Structure-activity relationship (SAR) studies at this position, driven by in silico docking and subsequent Cyp binding assessment by surface plasmon resonance (SPR), led us to TWH106, which bears a nitrophenyl group at position 3. TWH106 adopts a similar binding mode to SfA (PDB 1YND) and the previously reported macrocycles (PDB 6X4N) when docked into CypA. The nitrophenyl is expected to reach the K82/E81 sub-pocket in CypA and form a hydrogen bond with K82 (Fig. 1a). SPR with covalently immobilised CypA demonstrated that TWH106 binds CypA with similar affinity to CsA ($K_D$ 53 nM vs 67 nM) (Table 1 and Supplementary Fig. 2). We also measured the affinity of TWH106 to a closely related Cyp, endoplasmic reticulum (ER) located CypB, which is also involved in viral infection[54,55]. CypA and CypB have highly related core sequences, but CypB bears longer N- and C-termini which localise CypB to the ER (Supplementary Fig. 8c). We used a truncated version of CypB because the N-terminal 25 amino acid extension is proteolytically cleaved during recombinant expression in E. coli[56]. We found that TWH106 bound to this N-truncated CypB but with 2-3 fold lower affinity ($K_D$ 139 nM) compared to CypA (Table 1 and Supplementary Fig. 3).

TWH106 contains two chemically modifiable, solvent-exposed anchoring options for PROTAC linker attachment, methyls * and # (Fig. 1). Position # corresponds to the solvent-exposed mTyr residue in SfA, while position * coincides with the aliphatic spirocyclic extension

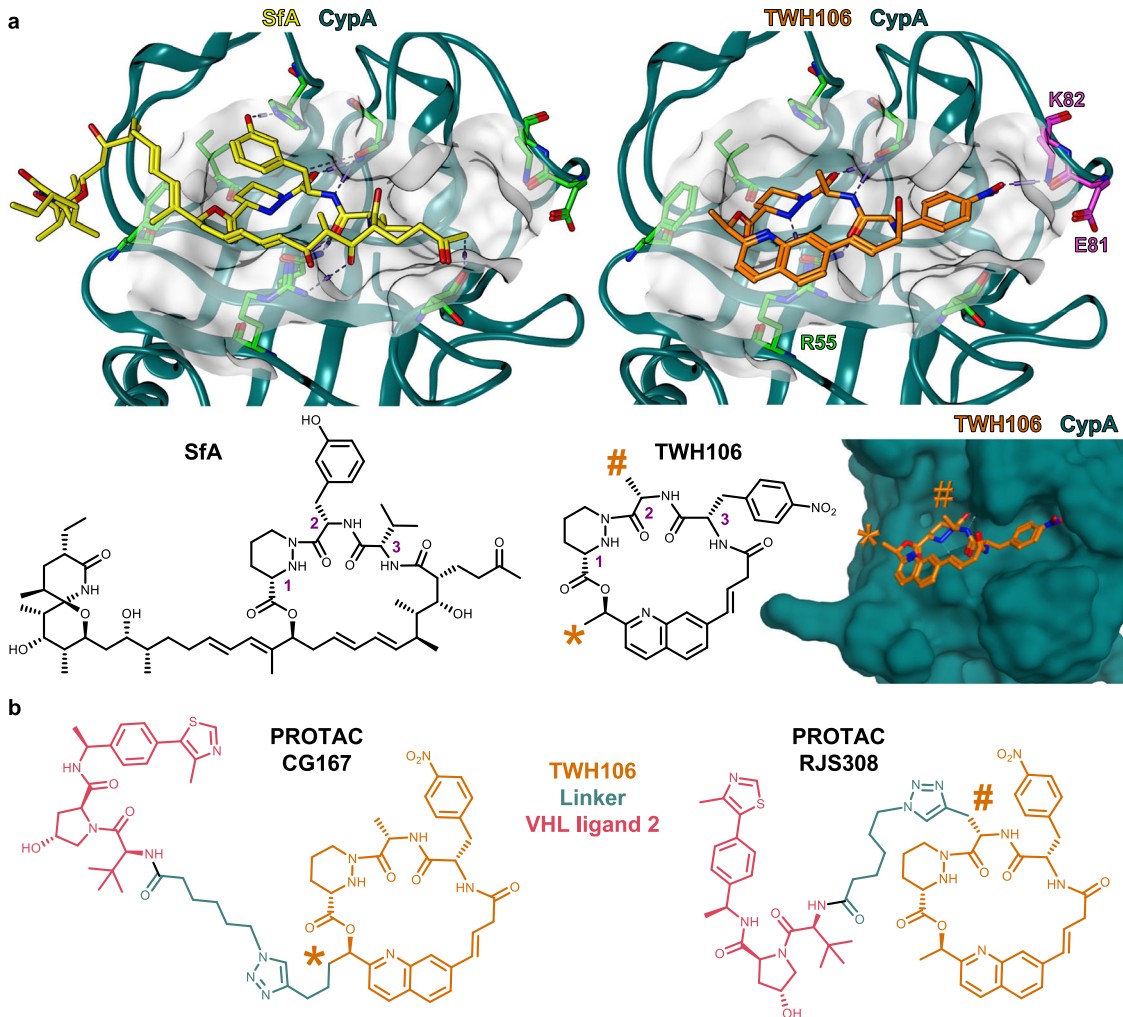

**Fig. 1 | Design of macrocycle-based PROTACs with high affinity for cyclophilins. a** X-ray crystal structure of sanglifehrin A-CypA complex (PDB 1YND) (left) and TWH106 docked into CypA (right). The TWH106 nitrophenyl moiety fills a subpocket near residues K82 and E81 and the nitro group H-bonds with K82 (arginine in CypB). Two solvent-exposed methyl groups ✳ and # of TWH106 were used to synthetically grow PROTACs. **b** Chemical structure of our two TWH106-based PROTACs CG167 and RJS308.

of SfA, which confers immunomodulatory activity via interaction with IMPDH2[44,47]. Therefore, we reasoned that linker attachment to either of these two positions should not impair PROTAC-CypA binding. For both CG167 and RJS308, the position of the linker with respect to the CypA active site is considerably different to that of our CsA-based PROTAC JW4-10 (Supplementary Fig. 10a), reported in Colpitts et al. 2020[21] (there termed CsA-Prtc1), which extends from the opposite corner of the CypA active site (Supplementary Fig. 8b).

### Table 1 | Affinity of Cyp inhibitors against CypA and CypB by SPR

|  | TWH106 | CG167 | RJS308 | CsA |
|---|---|---|---|---|
| CypA $K_D$ (nM) | 53 ± 5 | 64 ± 12 | 108 ± 12 | 67 ± 1 |
| CypB $K_D$ (nM) | 139 ± 24 | 71 ± 13 | 320 ± 40 | nd |

Affinities of TWH106 (non-PROTAC parental ligand), CG167 (Cyp-PROTAC) and RJS308 (Cyp-PROTAC) measured by SPR equilibrium analysis on an amide coupling CM5 chip derivatised with full-length CypA or N-truncated CypB. Kinetic analysis provided similar results. SPR curves and affinity plots for one representative experiment are shown in Supplementary Fig. 2 and Supplementary Fig. 3. Experiments were run in duplicate, and $K_D$s are means from two independent experiments ± SD. The affinity of CsA against CypA was also determined (*n* = 2), but not against CypB due to slow dissociation and the lack of successful regeneration strategies.

We previously found that our VHL-directed CsA-based PROTAC JW4-10 was a more effective CypA degrader in cells than its cereblon and MDM2-directed counterparts[57]. The von Hippel-Lindau (VHL) E3 ligase PROTAC ligands are also among the best-characterised and most successfully utilised[58]. We thus chose VHL-ligand 2 (Me-VH032) (Fig. 1b) as the E3 ligase ligand, a nanomolar VHL binder with optimised physicochemical properties that can be conveniently reacted at its amine moiety via amide coupling[59]. Using similar spacing between the buried macrocycle and E3 ligase ligand to previously reported JW4-10[21] (Supplementary Fig. 10a), we chose 10-12 atom long aliphatic linkers containing a triazole for final PROTAC assembly via robust and tolerant CuAAc chemistry[60]. This led to Cyp-PROTACs CG167 (linker position ✳) and RJS308 (linker position #) (Fig. 1b). Both PROTACs retained high affinity by SPR for CypA ($K_D$ 64 nM for CG167 and 108 nM for RJS308) and N-truncated CypB ($K_D$ 71 nM for CG167 and 320 nM for RJS308) (Table 1 and Supplementary Figs. 2 and 3).

**Synthesis of enantiopure macrocycle-based Cyp-PROTACs**
Our synthetic approach towards the TWH106 macrocycle requires access to intermediate **7**, which can be subsequently macrocyclised in a key intramolecular Heck cross-coupling reaction (Fig. 2). Intermediate **7** was obtained by a convergent route which assembles three building blocks: (*R*)-bromoquinoline alcohol **2**, Boc-protected

## Convergent synthesis of 7

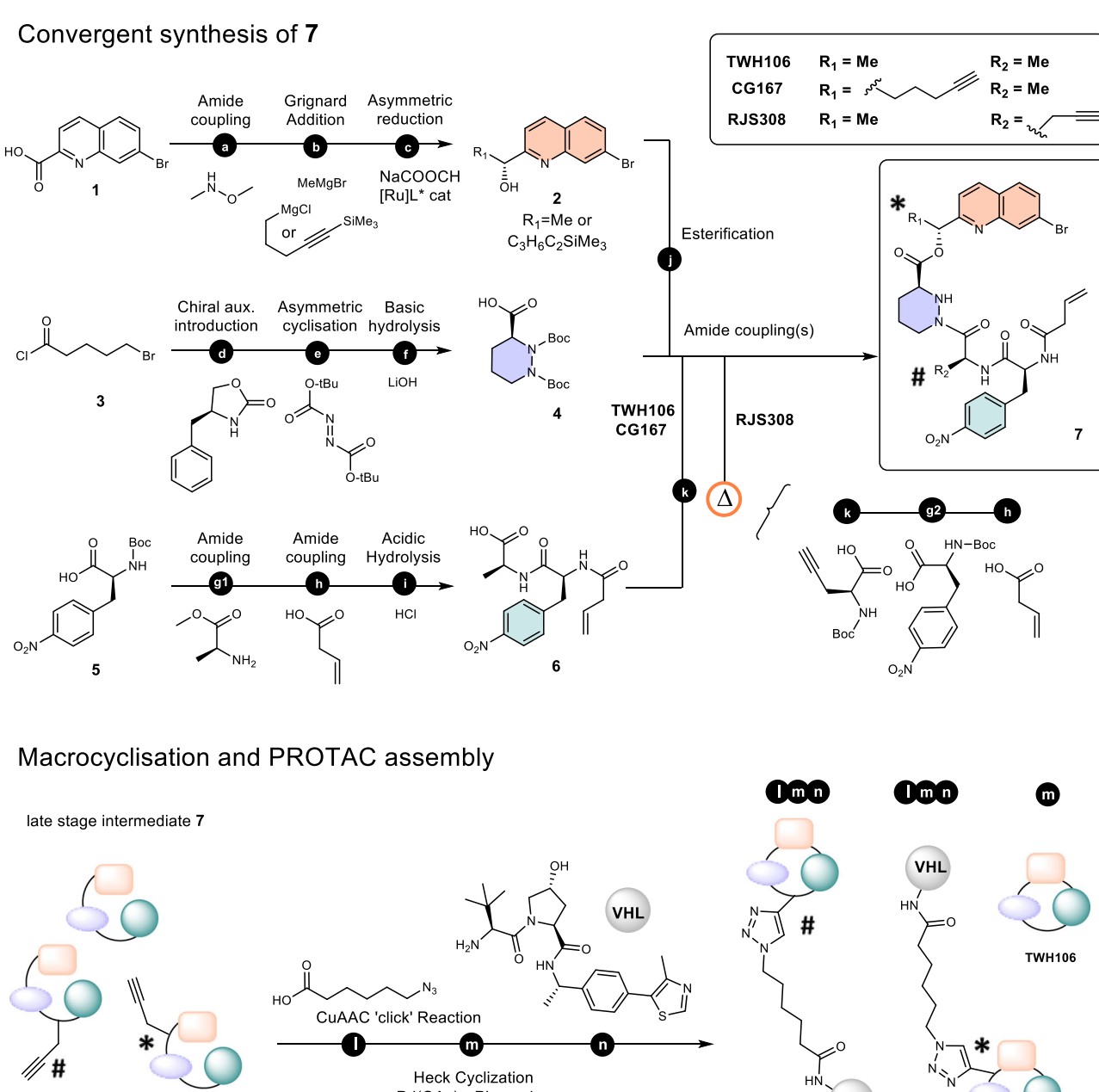

## Macrocyclisation and PROTAC assembly

**Fig. 2 | Synthesis of enantiopure macrocycle-based PROTACs. a** HATU, *i*-Pr₂NEt, DMF, 96% **b** THF, 97% or 28% (CG167) **c** dichloro(p-cy)Ru(II) dimer, (1 R, 2 R)–(−)-N-p-tosyl−1,2-diphenylethylenediamine, THF, H₂O, HCOONa, 90% or 86% (CG167) **d** n-BuLi, THF, 90% **e** LDA, THF, DMPU **f** THF, H₂O, 64% (2 steps) **g1** HATU, OxymaPure, *i*-Pr₂NEt, MeCN, 52% **g2** i. TFA, CH₂Cl₂ ii. Like G1, 69% (RJS308) **h** i. TFA, CH₂Cl₂ ii. PyAOP, *i*-Pr₂Net, CH₃CN, 41%, 57% (RJS308) **i** 2 M HCl, dioxane, H₂O, 66% **j** EDC,

DMAP, *i*-Pr₂NEt, CH₂Cl₂, 78% or 95% (CG167) **k** i. TFA, CH₂Cl₂ ii. HATU, *i*-Pr₂NEt, OxymaPure, 55% (TWH106), 38% (CG167), 54% (RJS308) **l** CuSO₄, L-ascorbic acid, THF, H₂O, 91% (RJS308), 82% (CG167) **m** Pd(OAc)₂, piperazine, DMF, μ-wave, 8% (TWH106), 27% (RJS308), 7% (CG167), **n** HATU, *i*-Pr₂NEt, DMF, 30% (RJS308), 80% (CG167). Characterisation data is provided in the Supplementary Information file.

piperazic acid **4** and tripeptide **6**. Briefly, bromoquinoline alcohol **2** can be accessed by reduction of a Weinreb amide with an alkyl Grignard, followed by enantioselective Noyori reduction to the (R)-alcohol, as described previously[51]. Enantiopure (3S)-piperazic acid **4** was synthesised using the procedure by Hale et al. with a chiral auxiliary[61]. Tripeptide **6** was synthesised through standard amine coupling strategies followed by slow acidic hydrolysis, which was key to preventing epimerisation of the amino acid stereocenters. Intermediate **7** was finally assembled by esterifying alcohol **2** with free piperazic acid **4** and coupling the tripeptide moiety **6** to the

deprotected pyridazine γ-nitrogen. In the case of RJS308, the commercial availability of the propargyl glycine starting material prompted us to build the peptide moiety on the esterified piperazic acid directly (Fig. 2 sequence Δ).

We chose alkynes as synthetic handles for PROTAC assembly on our highly functionalised macrocycle TWH106. Alkynes are robust enough to be introduced early but can be reacted late-stage in mild conditions and very specifically to triazoles via Huisgen click cycloaddition[62]. The alkyne synthetic handle for RJS308 (linker #) was introduced during peptide coupling with unnatural amino acid

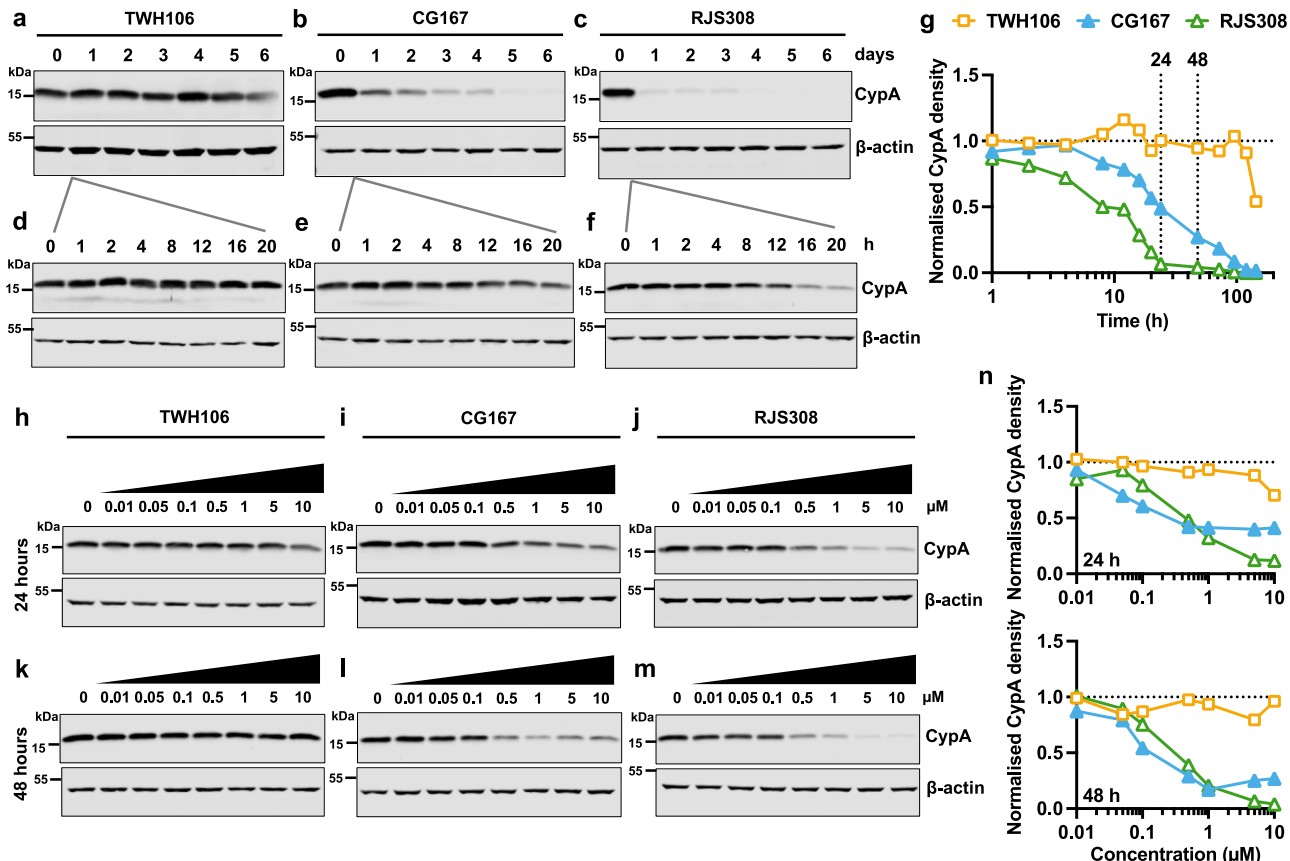

**Fig. 3 | PROTACs CG167 and RJS308 degrade CypA in a dose-dependent manner. a–f** Immunoblots detecting CypA, or β-actin loading control, in Jurkat cells treated with 5 μM TWH106, CG167 or RJS308 for up to 6 days (**a–c**) or 20 h (**d–f**). **g** CypA densities from (**a–f**) adjusted for loading by reference to β-actin densities, mean (*n* = 2 independent experiments). **h–m** Immunoblots detecting CypA in Jurkat cells treated with 0.01 – 10 μM TWH106, CG167 or RJS308 for 24 h (**h–j**) or 48 h (**k–m**). **n** CypA densities relative to β-actin (**h–j**) top and (**k–m**) bottom, mean (*n* = 2 independent experiments). Additional immunoblots in Supplementary Fig. 5. Source data are provided in the Source Data file.

L-propargylglycine (Fig. 2 step **Δ**k). The terminal alkyne for CG167 (linker *) was introduced by forming an alkyl Grignard from a silylated chloropentyne building block, which reacted with the Weinreb amide to produce the corresponding ketone (Fig. 2 step b). Noyori asymmetric reduction of the resulting alcohol remarkably retained its enantioselectivity, which was reflected in the high diastereoisomeric purity after reaction with (3S)-*bis*-Boc-piperazic acid as exemplified by NMR studies (>95:5 dr in supporting data compound **s11**). NOE analysis of CG167 is also in accordance with the proposed configuration of the stereocenter (Supplementary Fig. 4).

Unlike the previously reported macrocycles, which were synthesised by ring-closing metathesis, and require the introduction of an alkene substituent on the quinoline ring, we used a late-stage Heck cross-coupling to close our macrocycles directly. Due to observed cross-reactions of alkynes during our attempts at palladium-catalysed Heck cross-coupling reactions, the terminal alkynes were coupled first with azide linkers via copper(I)-catalysed alkyne-azide cycloaddition (CuAAc) before performing the macrocyclisation. The challenging Heck cyclisation gave low yields due to low reactivity and difficulties during purification, arising from side product formation. The *trans* alkene products could nevertheless be separated and obtained successfully. The cyclised products then reacted smoothly at the free carboxylic acid of the linker with the VHL ligand via amide coupling to form Cyp-PROTACs RJS308 and CG167. We envisage that this synthetic strategy will be helpful to sanglifehrin chemistry in general as the introduction of the linker * corresponds to the alkyl spirocyclic extension of SfA and avoids the use of organo-tin agents[63,64].

## Characterisation of the degradation potential of PROTACs CG167 and RJS308

We first tested whether Cyp-PROTAC treatment (CG167 or RJS308) caused CypA degradation in Jurkat cells, a human T cell line. Treatment for up to 6 days with 5 μM CG167 or RJS308 decreased CypA protein to almost undetectable levels (Fig. 3a–g). RJS308 treatment almost completely degraded CypA within 24 h, whereas CG167 activity was slower, requiring 4-5 days for a similar degree of CypA degradation (Fig. 3g). The parental Cyp ligand TWH106 did not reduce CypA levels except a small decrease after 6 days of continuous treatment (Fig. 3a, d, g). Cyp-PROTACs, but not TWH106, caused dose-dependent CypA degradation after 24 h and 48 h treatments (Fig. 3h–n). CypA DC$_{50}$s for the 48 h treatments were 284 nM (RJS308) and 123 nM (CG167) with D$_{max}$s of 98% (RJS308) and 76% (CG167) (means of *n* = 2 independent experiments), calculated from sigmoidal fits of immunoblot densities. For all experiments, normalised CypA densities were calculated from two independent experiments (additional immunoblots in Supplementary Fig. 5). High PROTAC concentrations can sometimes reduce activity through titration of the ternary complex, the so-called hook effect, but we did not observe this. This is perhaps linked to the high abundance of CypA in cells[65].

## CG167 and RJS308 degrade CypA via a PROTAC mechanism

We next demonstrated that CypA degradation was neddylation dependent for both Cyp-PROTACs as expected. NEDD8-activating enzyme inhibitor MLN4924 prevents the necessary neddylation of the active E3 ligase complex (Cul2-Rbx1-elonginB/C-VHL), which catalyses

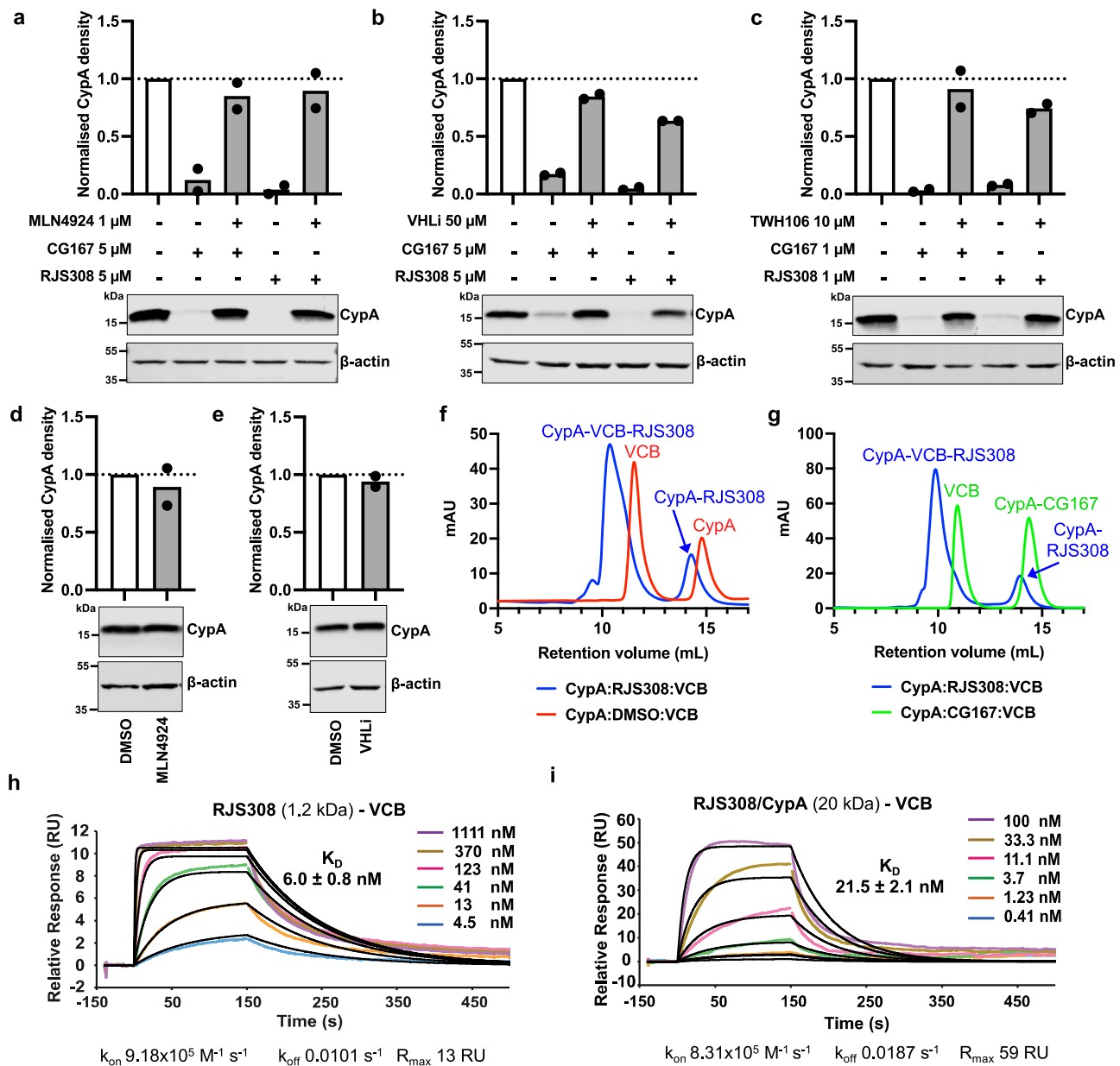

**Fig. 4 | Degradation of CypA by CG167 and RJS308 is via a PROTAC mechanism. a–e** Immunoblots detecting CypA, or β-actin as loading control, in Jurkat cells. Cells were pretreated with (**a**) 1 μM NEDD8-activating enzyme inhibitor MLN4924 for 6 h or (**b**) 50 μM VHL inhibitor VH298 (VHLi) for 2 h, followed by 5 μM PROTAC CG167 or RJS308 for 48 h, or cells were treated with (**c**) 10 μM TWH106 and 1 μM PROTAC CG167 or RJS308 for 48 h, or cells were treated with (**d**) MLN4924 as in (**a**) or with (**e**) VHLi as in (**b**). CypA densities (top) adjusted for loading by reference to β–actin densities, mean ($n$ = 2 independent experiments). Additional immunoblots in

Supplementary Fig. 6. **f, g** Size-exclusion chromatography traces for complex formation between VCB (pVHL/elongin C/elongin B, 41 kDa) and CypA (20 kDa) with (**f**) RJS308 (1.2 kDa, blue) vs DMSO (red) or (**g**) CG167 (1.2 kDa, green) vs RJS308 (1.2 kDa, blue) addition. **h, i** Dose-dependent binding between (**h**) RJS308 or (**i**) RJS308/CypA complex (flow) and VCB (immobilised on chip) in representative SPR experiments. Binding affinities were calculated from kinetic fits (black curves, 1:1 binding model), mean ± SD ($n$ = 5 (**h**) or $n$ = 4 (**i**)) independent experiments. Source data are provided in the Source Data file.

ubiquitination of the PROTAC target upstream of the proteasome. CypA levels were rescued after pretreatment with 1 μM MLN4924 for 6 h prior to Cyp-PROTAC treatment (5 μM, 48 h) (Fig. 4a, lanes 3 and 5). Importantly, MLN4924 had no impact on CypA levels in the absence of PROTAC (Fig. 4d). E3 ligase dependence was further evidenced by CypA rescue by a 2 h pretreatment with excess VHL inhibitor VH298 (VHLi)[66] (Fig. 4b, lanes 3 and 5). Reversal of CG167-mediated CypA degradation was more efficient, consistent with less effective degradation by CG167 (Fig. 3n). VHLi treatment did not impact CypA levels in the absence of PROTAC (Fig. 4e). Finally, PROTAC activity was also

evidenced by CypA rescue by an excess of the non-PROTAC parental Cyp inhibitor, TWH106 (Fig. 4c, lanes 3 and 5). For all experiments, CypA densities were calculated from two independent experiments (additional immunoblots in Supplementary Fig. 6). Together, these results evidence a ubiquitin-proteasome- PROTAC mechanism for CypA degradation by CG167 and RJS308.

We next sought biochemical evidence for ternary complex formation between CypA-PROTAC-VHL/elongin C/elongin B. Size exclusion chromatography (SEC) of mixtures of recombinant proteins and PROTACs indicated ternary complex formation with RJS308 (Fig. 4f).

Strikingly, we did not detect ternary complex formation with CG167, evidenced by the absence of the peak shift observed with isobaric PROTAC RJS308 (Fig. 4g). SPR experiments flowing RJS308, in the presence and absence of saturating CypA concentrations, on immobilised VCB complex (VHL/elongin C/elongin B) confirmed low nanomolar affinity of RJS308 to VCB alone and in complex with CypA (Fig. 4h, i). The considerable increase in $R_{max}$ (Fig. 4i) compared to RJS308 alone (Fig. 4h) further evidenced PROTAC-induced ternary complex formation. However, CG167 binding to VCB by SPR was not observed, even at the highest concentrations tested (10 µM) (Supplementary Fig. 7d). Fluorescence polarisation experiments further confirmed these results with a binary affinity of 104 nM between VCB and RJS308 while CG167 only began to display VCB binding at concentrations above 10 µM (Supplementary Fig. 7a–c). The reasons for the poorer binding of CG167 to VHL remain unclear, but this likely explains less efficient CypA degradation by CG167 as compared to RJS308. We note that despite the weak engagement with recombinant protein, some VHL engagement must occur in cells as CG167-mediated CypA degradation is prevented by the VHL inhibitor VH298 (Fig. 4b).

### PROTACs CG167 and RJS308 selectively degrade CypA

The *cis/trans* isomerase active sites of the 17 human cyclophilins are highly conserved[53] suggesting that active site-targeting inhibitors might be promiscuous. To examine PROTAC specificity, we used mass spectrometry-based proteomics to measure protein levels after 72 h treatment of THP-1 monocytic cells with 5 µM PROTAC or parental TWH106 (Fig. 5a–c). Strikingly, CypA was the most PROTAC-sensitive protein detected (Fig. 5b, c). As observed in previous experiments, TWH106 did not affect CypA levels (Fig. 5a). Other cyclophilin domain-containing proteins significantly reduced by PROTACs CG167 and RJS308, though to a lesser extent than CypA, included CypH and CypE, nuclear cyclophilins that participate in pre-mRNA splicing[67]. The peptides for related cyclophilin domain-containing proteins CypB, Cyp40, CypD, CypG, PPIL1, PPWD1, and Nup358 were all detected but showed no significant reduction. CypC peptides were not detected.

TWH106 design sought to exploit Cyp active site differences (Supplementary Fig. 8a), likely explaining differential sensitivity to isoforms through reducing Cyp binding affinity. Specificity for CypA over CypB, however, contradicts comparable CypB-PROTAC binding measured by SPR (Table 1). Ternary complex formation efficiency has been shown to govern PROTAC potency and selectivity[68–70], and Cyp surfaces do exhibit differential surface composition (Supplementary Fig. 8e, f). To investigate whether RJS308/CypB could form a ternary complex with VCB as observed with CypA (Fig. 4i), the SPR experiment was repeated with a RJS308/N-truncated-CypB complex (Fig. 5d). The slow dissociation constant of RJS308/CypB from VCB ($k_{off} < 0.002 \text{ s}^{-1}$) prevented accurate $K_D$ calculation due to the lack of appropriate regeneration conditions but revealed picomolar affinity, indicating that the globular domain of CypB is able to form a complex with VCB. We hypothesise that N- and C-termini flanking the different cellular Cyps (Supplementary Fig. 8c) likely influence PROTAC binding and/or ternary complex formation and explain specificity, for example for CypA over CypB. Subcellular location may also play a role (Supplementary Fig. 8d).

To probe CypA/CypB specificity further, we treated Jurkat cells with 5 µM TWH106, CG167 or RJS308 for 6 days and detected CypB daily by immunoblot (Fig. 5e–h and Supplementary Fig. 9a–c). Unexpectedly, CypB levels were most effectively reduced after treatment with the non-PROTAC Cyp ligand TWH106. TWH106-mediated CypB reduction was also observed by proteomics (Fig. 5a), although to a lesser extent. Regardless, PROTACs CG167 and RJS308 caused much less CypB degradation, with significant CypB loss requiring 4-5 days treatment with 5 µM PROTAC. Normalised CypB densities were calculated from two independent experiments (Supplementary Fig. 9a–c). Dose-response experiments revealed a dose-dependent

decrease in CypB after TWH106 treatment for 48 h whereas Cyp-PROTACs did not reduce CypB under these conditions (Fig. 5i–l and Supplementary Fig. 9d–f). Similar dose dependence was also visible after 24 h treatment with TWH106 (Supplementary Fig. 9g–m). The slow effect of PROTACs CG167 and RJS308 on CypB, combined with CypB loss after treatment with non-PROTAC TWH106, suggests more complex PROTAC-independent mechanisms of CypB regulation.

Our previously described VHL-directed CsA-based PROTAC (JW4-10)[21] degraded both CypA and CypB in Jurkat cells after 48 h (Supplementary Fig. 10). We have previously shown that JW4-10-mediated CypA degradation is sensitive to proteasome inhibitor MG132[21]. Here, looking at both CypA and CypB degradation, we found that pretreatment of Jurkat cells with NEDD8-activating enzyme inhibitor MLN4924 (1 µM, 6 h) prior to treatment with 5 µM JW4-10 fully rescued CypA, but not CypB (Supplementary Fig. 10b). Analogously, pretreatment with excess VHL inhibitor VH298 (VHLi) rescued CypA, but not CypB, from JW4-10 (Supplementary Fig. 10c). These findings suggest that the JW4-10-mediated decrease in CypB levels is VHL-independent and does not involve a PROTAC mechanism.

In conclusion, the development of TWH106 into PROTAC molecules improved their selectivity, reducing CypB loss observed with TWH106 treatment. Unlike CsA-based PROTAC JW4-10, neither CG167 nor RJS308 deplete CypB when used at concentrations that effectively degrade CypA. Indeed, PROTACs CG167 and RJS308 display striking specificity for CypA over closely related cyclophilins and cyclophilin-like proteins.

### Selective degradation of CypA over CypB is consistent across cell lines and primary cells

We next evaluated Cyp-PROTAC activity against CypA and CypB in a variety of cell lines and primary human cells. We measured protein levels by immunoblot after 48 h treatment with 5 µM CG167 or RJS308 in the following cells: Calu-3 airway cell line, THP-1 monocytic cell line, primary human monocyte-derived macrophages (MDMs), U87 glioma cell line, Huh7 hepatoma cell line, Jurkat T cell line, and primary human CD4+ T cells that were resting or activated (Supplementary Fig. 11a–g). Normalised densities of CypA and CypB relative to β-actin (Fig. 5m, n and Supplementary Fig. 11h) were calculated from two independent experiments (Supplementary Fig. 11a–g). CypA protein levels were reduced following 5 µM treatment with either Cyp-PROTAC in all cell types, in some cases to almost undetectable levels. Densitometry levels for CG167; RJS308 were: Calu-3 (65%; 39%), THP-1 (4%; 1%), MDMs (6%; 1%), U87 (51%; 19%), Huh7 (16%; 9%), Jurkat (13%; 4%), resting T cells (43%; 23%) and activated T cells (47%; 24%). RJS308 treatment degraded CypA more effectively than CG167 in all cell types, consistent with measurements of PROTAC ternary complex formation efficiency (Fig. 4f–i and Supplementary Fig. 7). Importantly, selectivity for CypA over CypB was preserved across cell types (Fig. 5m, n and Supplementary Fig. 11h).

### Cyp-PROTACs and parental Cyp ligand inhibit HIV-1 infection

Human immunodeficiency virus type 1 (HIV-1) uses CypA to evade the innate immune sensor and antiviral protein TRIM5α[23,25]. Thus, CypA inhibitors suppress HIV-1 replication with efficacy depending on TRIM5α activity in the infected cells. HIV-1 is, therefore, an excellent model for Cyp-PROTAC antiviral testing. We tested Cyp-PROTACs in U87 glioma cells, which express active TRIM5α[24]. We pretreated cells with Cyp-PROTACs or TWH106 for 48 h before infection with VSV-G pseudotyped HIV-1 lentiviral vector encoding GFP (HIV-1 GFP) reading out infection by counting GFP positive cells at 48 h post-infection (Supplementary Fig. 12a, b). Cell viability was measured in parallel by MTT assay demonstrating that TWH106 was slightly toxic at 10 µM while the Cyp-PROTACs were not (Supplementary Fig. 12e). All three Cyp inhibitors exhibited antiviral activity, but Cyp-PROTACs CG167 and RJS308 were less effective than parental Cyp

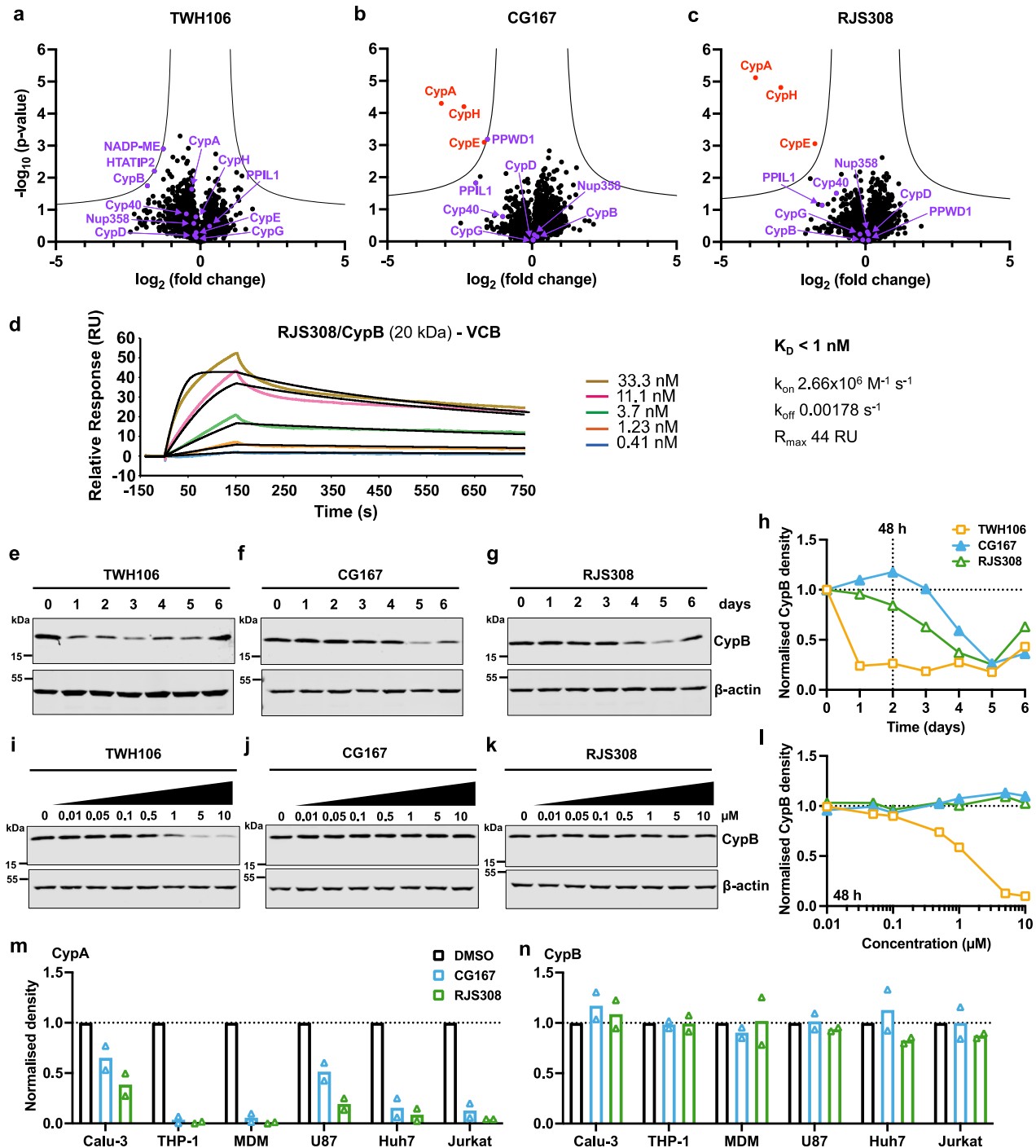

**Fig. 5 | Cyp-PROTACs are selective for CypA. a–c** Volcano plots of THP-1 cell proteomics analysis after 72 h treatment with 5 µM (**a**) TWH106 (parental Cyp ligand), (**b**) CG167 (Cyp-PROTAC), or (**c**) RJS308 (Cyp-PROTAC), compared to DMSO control, significant change (red), other proteins of interest (purple). Two-sided two-sample student t-test, s0 = 0.5; FDR = 0.05 (n = 3 independent experiments). **d** Dose-dependent binding (SPR) of RJS308/N-truncated CypB complex (flow) and VCB (immobilised on chip) with kinetic fits (black curves, 1:1 binding model) (n = 1). Slow complex dissociation indicates sub-nanomolar affinity but prevents accurate binding affinity determination. **e–g**, **i–k** Immunoblots detecting

CypB, or β-actin as loading control, in Jurkat cells treated with TWH106, CG167 or RJS308 at (**e–g**) 5 µM for 6 days or (**i–k**) 0.01–10 µM for 48 h. **h**, **l** CypB densities (**h**) from (**e–g**), (**l**) from (**i–k**) adjusted for loading by reference to β–actin densities, mean (n = 2 independent experiments). **m**, **n** CypA densities (**m**) or CypB densities (**n**) relative to β-actin from immunoblots after 48 h treatment with 5 µM CG167 or RJS308 on cells, mean (n = 2 independent experiments). Additional immunoblots for (**e–l**) in Supplementary Fig. 9, and immunoblots for (**m**, **n**) in Supplementary Fig. 11. Source data are provided in the Source Data file.

ligand TWH106 in this experiment (Supplementary Fig. 12b). We then tested anti-HIV-1 activity in Jurkat T cells, which exhibit better sensitivity to PROTAC-driven CypA degradation (Fig. 5m and Supplementary Fig. 11d, f). Jurkat cells were pretreated for 48 h before infection with an HIV-1 ΔEnv construct, in which the envelope is deleted and GFP

is expressed in place of the *nef* gene (HIV-1 LAI ΔEnv GFP). Despite the effective degradation of CypA in this experiment, measured by flow cytometry (Supplementary Fig. 12d), Cyp-PROTAC potency against HIV-1 was only slightly improved compared to U87 cells (Supplementary Fig. 12c). Notably, TWH106 was toxic to Jurkat cells at the highest

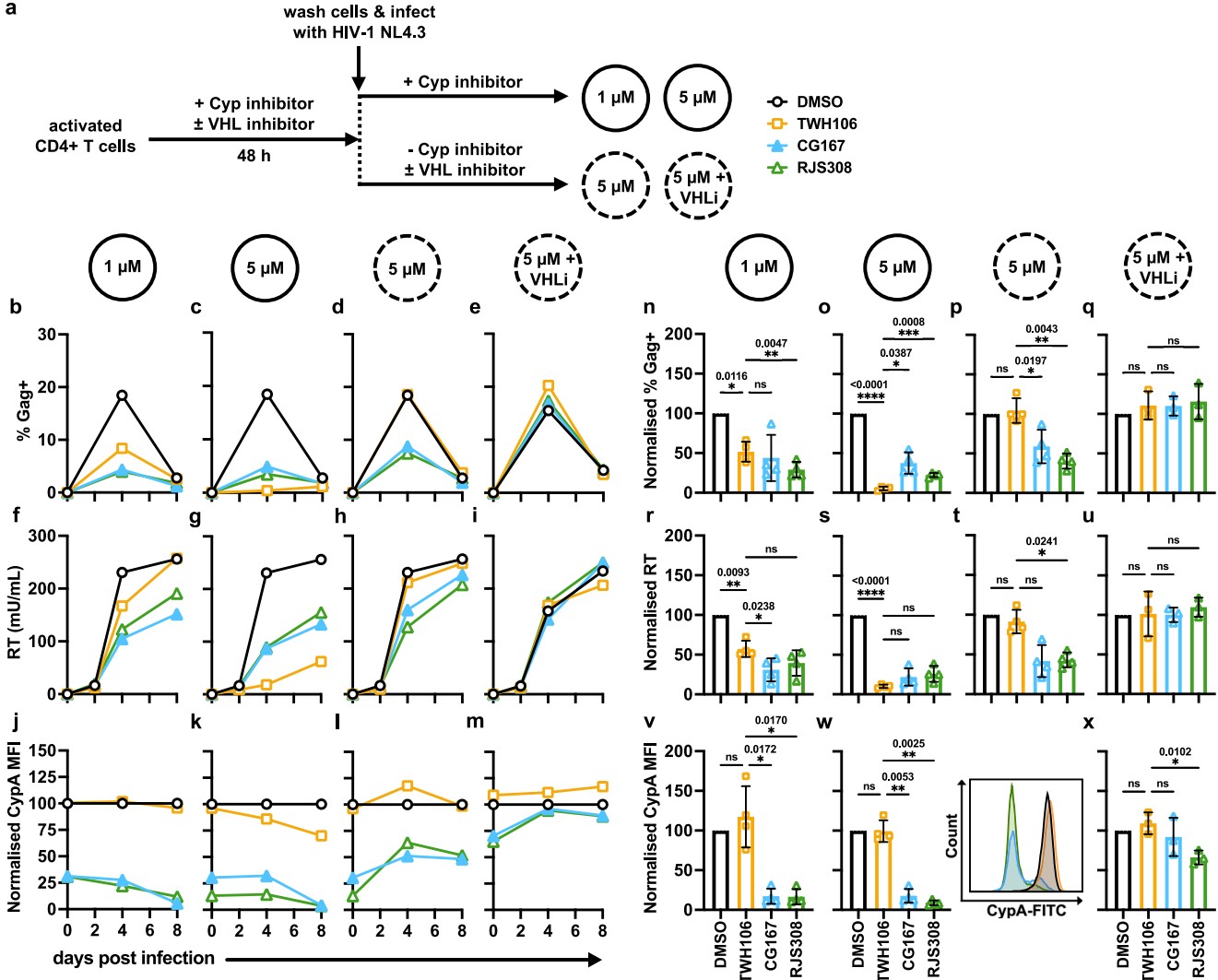

**Fig. 6 | Cyp-PROTACs exhibit improved anti-HIV-1 activity compared to parental Cyp ligand TWH106. a** Experimental design for (**b–x**): activated primary CD4+ T cells were pretreated for 48 h with TWH106 (orange), CG167 (blue), RJS308 (green) or DMSO (black), washed and infected with HIV-1 NL4.3 (2000 mU RT/10$^6$ cells). Cells were treated with 1 μM or 5 μM Cyp inhibitor throughout the experiment (solid circles) or only pretreated with 5 μM Cyp inhibitor (dashed circle). Alternatively, cells were pretreated with 5 μM Cyp inhibitor and 50 μM VHL inhibitor VH298 (5 μM + VHLi, dashed circle). **b–m** Representative data from one donor at indicated days post infection (dpi), mean (*n* = 1 independent experiment performed in duplicate), (**b–e**) % Gag+ cells (**f–i**) virus levels in supernatant measured by SG-PERT (**j–m**) CypA mean fluorescence intensity (MFI), normalised to

DMSO. **n–x** Data from 4 donors (1 μM, 5 μM and 5 μM pretreatment only) or 3 donors (5 μM pretreatment only + VHL inhibitor), all data normalised to DMSO, mean ± SD, (**n–q**) % Gag+ cells at 4 dpi, (**r–u**) virus levels in supernatant at 4 dpi measured by SG-PERT, (**v–x**) CypA MFI at time of infection (0 dpi). Statistical comparison using RM one-way ANOVA with Dunnett's multiple comparisons test comparing to TWH106, * (*P* ≤ 0.05), ** (*P* ≤ 0.01), *** (*P* ≤ 0.001), **** (*P* ≤ 0.0001), *P*-values shown. Data from additional donors in Supplementary Fig. 14. **w** (right) histogram showing the spread of CypA-FITC fluorescence at 0 dpi, representative data from one donor. Gating strategies are shown in Supplementary Fig. 15a–c. Source data are provided in the Source Data file.

concentrations tested, but Cyp-PROTACs were not (Supplementary Fig. 12f).

## Cyp-PROTACs exhibit improved anti-HIV-1 activity compared to parental Cyp ligand TWH106

Viruses are most dependent on cofactor interactions in the cells they naturally infect in vivo. In primary cells, TRIM5 activity is greater[23], and inhibitors are typically more effective against wild-type HIV-1 replication than single-round infection. Therefore, we measured Cyp-PROTAC anti-HIV activity against wild-type HIV-1 replication in primary human activated CD4+ T cells (Fig. 6). Activated T cells were treated with 1 μM or 5 μM Cyp inhibitor (TWH106, CG167 or RJS308) for 48 h, washed and infected with HIV-1 molecular clone NL4.3 (Fig. 6a). Cyp inhibitors or DMSO were readded after infection (1 μM or 5 μM, solid circles), or not readded (5 μM, dashed circle). Alternatively,

cells were pretreated with 5 μM Cyp inhibitor and 50 μM VHL inhibitor VH298 (5 μM + VHLi, dashed circle) (Fig. 6a). Infection levels (Fig. 6b–e, n–q) and CypA levels (Fig. 6j–m, v–x) were measured by fluorescent staining of cells for Gag or CypA, respectively, followed by flow cytometry analysis. In addition, viral particle release was determined by SG-PERT measurement of viral reverse transcriptase (RT) in the culture supernatants (Fig. 6f–i, r–u).

Figure 6 shows representative data for one donor over 8 days post infection (dpi) (Fig. 6b–m) (with additional data for this donor in Supplementary Fig. 13d) and normalised data from all donors at 4 dpi (% Gag +, RT) or 0 dpi (CypA mean fluorescence intensity (MFI)) (Fig. 6n–x). Supplementary Fig. 14 shows timecourse data for all T cell donors. Continuous treatment with either 1 μM or 5 μM Cyp inhibitor demonstrated that TWH106, CG167 and RJS308 were antiviral, evidenced by reduced numbers of Gag+ cells (Fig. 6b, c, n, o), and

reduced viral particles in culture supernatants (Fig. 6f, g, r, s), compared to DMSO-treated cells. Notably, at late time points (8 dpi), Gag expression measured by flow cytometry was suppressed as a result of cell death, leading to similarly low % Gag+ measurements. However, for measurements of infectious virus in the supernatant (RT), differential antiviral activity was observed at these later time points. Importantly, Cyp-PROTACs, but not parental Cyp ligand TWH106, led to effective degradation of CypA at both concentrations, shown by reduced CypA MFI (Fig. 6j, k, v, w). CypA levels at 0 dpi across all donors did not differ significantly between 1 μM and 5 μM treatments (Supplementary Fig. 13c). The Cyp-PROTACs, CG167 and RJS308, were more antiviral than TWH106 at 1 μM treatment (Fig. 6b, f, n, r), but less antiviral than TWH106 at 5 μM treatment (Fig. 6c, g, o, s). Indeed, the antiviral potency of the Cyp-PROTACs at 4 dpi was not significantly different between 1 μM and 5 μM, while at 1 μM TWH106 was significantly less antiviral compared to 5 μM (Supplementary Fig. 13a, b). These data illustrate the greater potency of Cyp-PROTACs at lower concentrations compared to TWH106. CG167 was slightly toxic at 5 μM (and 1 μM for some donors), decreasing the number of live cells (% of all cells) compared to DMSO-treated cells (Supplementary Figs. 13d and 14). These cells were excluded from analyses by gating on live cells (Supplementary Fig. 15b).

To further demonstrate that Cyp-PROTAC antiviral activity is indeed linked to CypA degradation, we carried out an analogous experiment where pretreated Cyp inhibitors were not readded to cells after washing and prior to infection (5 μM, dashed circle). As expected, CypA levels in Cyp-PROTAC-treated cells increased after Cyp-PROTAC withdrawal, indicating CypA protein resynthesis (Fig. 6l). TWH106 antiviral activity was abrogated in this experiment, whereas Cyp-PROTACs still retained some antiviral activity (Fig. 6d, h, p, t). These data emphasised the role of CypA degradation in Cyp-PROTAC antiviral activity, which was further evidenced by cotreatment with VHL inhibitor VH298 (VHLi) (5 μM + VHLi, dashed circle) in the pretreatment-only experiment, which rescued both CypA protein (Fig. 6m, x) and viral replication (Fig. 6e, i, q, u), demonstrating that Cyp-PROTAC antiviral activity is VHL dependent.

Together, these data illustrate that Cyp-PROTAC antiviral activity correlates with CypA depletion and that when inhibitor TWH106 is limiting, PROTAC-mediated degradation of CypA is more effective than competitive inhibition by TWH106. Furthermore, these data suggest that CypA degradation is most effective in wild-type HIV-1 replication models. Note that VHL protein levels in U87 cells, T cells and Jurkat cells do not account for differential PROTAC activity against CypA across these cells, since immunoblotting showed VHL was expressed at a lower level in primary T cells, the cell type in which PROTACs were most effective antivirals (Supplementary Fig. 12g).

## Cyp-PROTACs exhibit improved anti-HCV activity compared to parental Cyp ligand TWH106

HCV also depends on CypA as a viral cofactor, so we next tested our Cyp inhibitors against HCV replicon in a hepatoma cell line (Supplementary Fig. 16). Huh7 cells were electroporated with HCV JFH-1 replicon RNA (also encoding a luciferase gene) and treated with titrated concentrations of TWH106 or Cyp-PROTACs. All Cyp inhibitors were antiviral, evidenced by the reduction in luciferase at 48 h post electroporation (hpe) compared to DMSO-treated cells (Supplementary Fig. 16a). However, Cyp-PROTACs CG167 and RJS308 were similarly potent to parental Cyp ligand TWH106 at low concentrations and less potent at high concentrations. Cyp inhibitors were not toxic at any concentration tested (Supplementary Fig. 16b). To better characterise PROTAC antiviral activity and give time for CypA degradation we next carried out experiments where cells were pretreated with 1 μM or 5 μM Cyp inhibitor for 48 h and infected with HCV RNA before Cyp inhibitors were readded (Supplementary Fig. 16c–f). Cyp-PROTACs led to effective degradation of CypA at both 1 μM and 5 μM (Supplementary

Fig. 16e, f) and were potently antiviral at both concentrations (Supplementary Fig. 16d). Importantly, at 1 μM Cyp-PROTACs were more potent than TWH106, indicating the superiority of CypA degradation compared to inhibition at low concentrations, as observed in our HIV-1 spreading infection experiments.

To further demonstrate the role of CypA degradation in PROTAC antiviral activity, we carried out an experiment where pretreated Cyp inhibitors were washed out prior to electroporation and not readded, or cells were pretreated with 5 μM Cyp inhibitor and 50 μM VHL inhibitor VH298 (VHLi) before electroporation and not readded (Supplementary Fig. 16c, g–j). These treatment conditions were analogous to those for HIV-1 spreading infection experiments in primary T cells (Fig. 6a), and results were comparable, with TWH106 antiviral activity being abrogated in this experiment, while Cyp-PROTACs retained antiviral activity (Supplementary Fig. 16g, plain bars). The addition of VHLi partially rescued replication (Supplementary Fig. 16g, striped bars) and CypA levels (Supplementary Fig. 16i, j), demonstrating VHL dependency of PROTAC antiviral activity. Neither the Cyp inhibitors nor VHLi were toxic (Supplementary Fig. 16h).

These data further demonstrate the improved antiviral activity of Cyp-PROTACs compared to competitive inhibition, illustrating the benefit of the PROTAC approach.

## Discussion

Here, we demonstrate a 14-step convergent synthesis to access PRO-TAC designs based on TWH106, a natural product-derived macrocycle. We accessed PROTACs by derivatisation at the tripeptidic unit of SfA and at the SfA arm originally corresponding to the spirocyclic extension that confers immunosuppressive activity. Our Cyp-PROTACs retained low nanomolar affinity against target cyclophilin protein CypA compared to the parental TWH106 (Table 1). Critically, our Cyp-PROTACs are based on non-immunosuppressive, pharmacokinetically optimised macrocycles unrelated to CsA[51,52], which present fewer off-target effects than CsA-based molecules[49]. We have shown that both Cyp-PROTACs CG167 and RJS308 degrade CypA via a PROTAC mechanism involving VHL recruitment (Fig. 4a–c). While VHL engagement in cells is demonstrated by the rescue of CG167-mediated CypA degradation with VH298 (VHLi) (Fig. 4b), the slower kinetics of CG167 in cells (Fig. 3a–g) is consistent with reduced VHL engagement and poor ternary complex formation measured biochemically (Fig. 4g). The molecular basis for this is still unclear, as RJS308 and CG167 have identical attachment positions to the VHL binding moiety and differ only slightly by linker structure. Certainly, previous studies have shown that small changes in PROTAC linker can drastically affect E3 ligase affinity[71].

Proteomic experiments revealed that CypA is the most sensitive protein to PROTACs CG167 and RJS308 (Fig. 5b, c). Related cyclophilins CypH and CypE were decreased but CypA degradation was dominant.

The differential binding affinity of the warhead to Cyps due to sequence changes in the nitrophenyl binding region (Supplementary Fig. 8a) could explain some of this selectivity. Protein sequences outside the Cyp domain are also expected to have an important role in defining Cyp-PROTAC specificity. Cyp domains are typically flanked by divergent N- and C-termini (Supplementary Fig. 8c), which likely influence their conformational dynamics, impacting PROTAC recruitment and ternary complex formation as well as their cellular location. This is supported by SPR experiments with CypA and a truncated CypB, which allowed us to compare specificity for the Cyp domains. Our PROTACs had comparable affinities for both CypA and CypB (Table 1), indicating that specificity is not conferred by the Cyp domain. Another consideration is that PROTACs, or their corresponding ligase cofactors, may poorly access specific regions of the cell (Supplementary Fig. 8d), for example, CypB in the ER[56].

Strikingly, prolonged treatment with Cyp-PROTAC CG167 or RJS308 caused CypB loss at longer time points (Fig. 5f–h). However,

the non-PROTAC parental inhibitor TWH106 caused faster CypB loss suggesting a PROTAC-independent mechanism for this particular target protein (Fig. 5e, h, i, l). Our previously reported CsA-based PROTAC JW4-10[21] mediated MLN4924 and VHLi insensitive CypB degradation, also consistent with a PROTAC-independent mechanism for CypB loss (Supplementary Fig. 10). CypB loss in this case may be explained by CypB secretion upon inhibitor binding, a phenomenon reported for CsA[72] and recently for SfA[73].

Our results highlight the additional layers of selectivity exerted by PROTACs in cellular systems compared to competitive inhibitors. In particular, the cleaner profile of PROTACs CG167 and RJS308 compared to other inhibitors could simplify the investigation of the roles of different cyclophilins in viral infections and beyond. For example, our TWH106 series may be useful to distinguish the effect of CypA inhibition at the PPIase active site versus protein removal. Notably, we previously found that treating HCV-infected cells with CsA leads to innate immune activation and interferon (IFN) production, whereas degradation of CypA with CsA-based PROTAC JW4-10 inhibited HCV without inducing IFN[21]. A CypA-targeting PROTAC has recently been described that counteracts the proinflammatory cytokine storm production following influenza B virus (IBV) infection in mice[74], concordant with CypA degradation preventing IFN induction during HCV infection. Cyp-PROTACs thus make excellent tools for further dissection of these complex host-virus interactions. In this study we have focused on CypA because it was most sensitive to PROTAC degradation in proteomic experiments (Fig. 5b, c). However, these experiments also revealed degradation of CypE and CypH. While CypA has an established role as a viral cofactor, CypE and CypH have also been suggested to potentially be involved in HCV replication[75]. We cannot rule out the possibility that degradation of these proteins contributes to the antiviral activity of our Cyp-PROTACs.

We demonstrate that our Cyp-PROTACs have antiviral activity against HIV-1 replication in primary human CD4+ T cells (Fig. 6) and against HCV replicon in Huh7 cells (Supplementary Fig. 16). We find increased antiviral activity of our PROTACs over non-PROTAC parental inhibitor TWH106 in specific circumstances, most importantly when inhibiting HIV-1 replication in primary T cells with lower inhibitor concentration (1 μM), or when the inhibition duration was reduced to pretreatment only. We hypothesise that at high concentration (5 μM) TWH106 is more effective than the PROTACs because the PROTAC modifications affect cell permeability and reduce target engagement. However, at lower concentrations (1 μM), when an inhibitor is limiting, a PROTAC catalytic degradation mechanism has improved antiviral activity, compared to active site competitive inhibition. Similarly, when cells are pretreated with inhibitors, HCV replicon replication is suppressed more effectively by non-PROTAC parental inhibitor TWH106 at a high dose (5 μM), while at a lower dose (1 μM) the PROTACs are superior. PROTAC-mediated degradation may also gain antiviral activity through loss of target function unrelated to the active site. We also note that TWH106 is a more effective inhibitor than the PROTACs in cell line models of HIV-1 (Supplementary Fig. 12a–f). While we do not understand what differs between the cell lines and the primary T cells, we assume that the primary T-cell experiments represent the situation in vivo better than the cell lines. Notably, it has previously been shown that the ability of CypA to protect HIV-1 from restriction by TRIM5α is more pronounced in primary cells[23].

These data generally support the development of PROTACs as antivirals, with PROTACs giving a particular advantage at lower doses or when exposure to inhibitors is reduced. Indeed, PROTACs are emerging as effective antivirals. Exemplars include a telaprevir-based cereblon E3 ligase PROTAC, which degrades NS3/4A, inhibits HCV infection in vitro, and strikingly rescues activity against telaprevir-resistant NS3 mutants[10]. Similarly, degradation of influenza neuraminidase with an oseltamivir PROTAC has potent anti-influenza activity, including against oseltamivir-resistant strains[76]. A series of indomethacin-based PROTACs has recently been reported that target SARS-CoV-2 main protease to VHL, and inhibit infection in human lung cells[77]. Another recent study describes the cereblon PROTAC-mediated degradation of the HIV-1 Nef protein, which rescues Nef-mediated MHC-I and CD4 downregulation, and inhibits HIV-1 replication in primary T cells[78]. In one host-targeting approach, a PROTAC based on an existing cyclin-dependent-kinase-9 (CDK-9) inhibitor has been developed with activity against human cytomegalovirus and SARS-CoV-2[79].

Our cyclophilin A degraders further illustrate how PROTACs can be synthetically tractable, specific degraders and effective inhibitors of viral replication. We also expect them to be valuable tools in probing the role of cyclophilin A in viral infection and beyond.

## Methods

### Compound synthesis and characterisation
Detailed descriptions of synthetic methods and compound characterisation, as well as structural NMR studies, are provided in the Supplementary Information file. For biological assays, compounds were dissolved in anhydrous DMSO (900645, Sigma Aldrich) and stored in aliquots at −20 °C with drying agent.

### Surface plasmon resonance (SPR) against CypA and CypB
Compounds were evaluated for their affinity towards cyclophilin A and B by SPR using a Biacore T200 system at 25 °C. Recombinant full-length human CypA protein with an N-terminal His$_6$ tag was a gift from Leo James. CypA was diluted to 50 μg/mL in 10 mM sodium acetate at pH 5. Commercial CypB, which has a truncated N-terminus (D34-A212, 11004-H08H-SIB), was diluted to 25 μg/mL in 5 mM maleate buffer at pH 7. To improve the directionality of the immobilisation, a binding site blocker (60 μM JW47[80], a bulky CsA analogue) was added to the diluted protein solutions. This mixture was allowed to stand at room temperature for 10 min prior to set-up in the injection rack. Data was collected on a dual flow cell on CM5 amine coupling sensor chips (Cytiva BR100530). The surface was initially activated with 1:1 N-hydroxysuccinimide (0.1 M) and 1-ethyl-3-(3-dimethylaminopropyl) carbodiimide (0.4 M) for 420 s at 10 μL/min. Protein solutions were pulsed on the surface, and the surface was then quenched by 420 s of ethanolamine (1 M, pH 8.0). The blank reference channel was prepared in the same way omitting the injection of protein. Chips with immobilisation between 3000 and 4000 RU were obtained. For CypB, the chip was washed for 2 h with buffer to dissociate JW47. Compounds were injected over the surface for 120 s and left to dissociate for 600 s or 500 s at a 30 μL/min flow rate. Experiments were repeated twice with independent chips, and all injections were run in duplicate, including a solvent correction. Sensorgrams from reference surfaces and blank injections were subtracted from the raw data before data analysis using Biacore T200 software. Equilibrium responses against concentration were plotted and analysed by non-linear regression (sigmoid fit) to obtain binding affinities using Origin (OriginLab, version 9.8.0.200).

### SPR against VCB (VHL/elongin C/elongin B)
Surface plasmon resonance experiments were performed on Biacore 8 K instruments (Cytiva). Biotinylated Avi-tagged VCB was immobilised into a streptavidin pre-coated SPR sensor (Cytiva BR100531) at 2000 RU. Biotinylated Avi-tagged VCB was prepared as previously described[68]. Excess-free biotin was removed by extensive dialysis (3-times) into a buffer containing 20 mM HEPES pH 7.5, 200 mM NaCl and 0.25 mM TCEP. All interaction experiments involving VCB were done at 20 °C in running buffer (20 mM HEPES, pH 8.0, 150 mM potassium chloride, 2 mM magnesium chloride, 2 mM TCEP, 0.05% Tween20, 2% DMSO). The PROTACs were diluted in a running buffer and injected over the immobilised target proteins (concentration range, 3.33 – 1000 nM). For ternary complex measurements, experiments were run in the presence of the saturating amount of CypA or CypB (1 μM)

during the injection phase. Recombinant human full-length tag-free CypA was obtained as previously described[81], and N-truncated CypB was obtained commercially (11004-H08H-SIB).

Data analysis was performed as described above. Sensorgrams from reference surfaces and blank injections were subtracted from the raw data before data analysis using Biacore Insight software. Affinity and binding kinetic parameters were determined by using a 1:1 interaction model.

## Size exclusion chromatography

The ability of Cyp-PROTACs to induce ternary complex was assessed by size exclusion chromatography using an AKTA Pure system (Cytiva). The recombinant VCB (pVHL-elongin C-elongin B) complex was expressed and purified as described previously[69]. Superdex S75 10/300 GL (Cytiva) was equilibrated with a buffer containing HEPES pH 7.5 (20 mM), NaCl (100 mM) and TCEP (1 mM). VCB, Cyp-PROTAC (RJS308 or CG167) and CypA were mixed in the ratio of 1:1:1.4 and left incubated on ice for 1 h prior to the column run.

## Fluorescence polarisation (FP)

The ability of Cyp-PROTACs to bind VHL was assessed by an FP displacement assay described previously[68]. All FP measurements were taken using a PHERAstar FSX (BMG LABTECH) with fluorescence excitation and emission wavelengths ($\lambda$) of 485 nm and 520 nm, respectively. Competitive binding assays were performed in triplicate on 384-well plates (#3575, Corning) with a total well volume of 15 μL. Each well solution contains FAM-labelled HIF-1α peptide JC9 (10 nM)[82], VCB protein (15 nM), and decreasing concentrations of Cyp-PROTAC or Cyp-PROTAC:CypA complex in buffer containing Bis-tris propane pH 7.0 (100 mM), sodium chloride (100 mM), TCEP (1 mM), with a final DMSO concentration of 1%. To obtain the percentage of displacement, control wells containing peptide in the absence of protein (maximum displacement, unbound peptide), or VCB and peptide with no compound (zero displacement) were also included. These values were then fitted by non-linear regression using Prism (GraphPad, version 7.03) to determine average $IC_{50}$ values. A displacement binding model was used to back-calculate inhibition constants ($K_D$) from the measured $IC_{50}$ values[82].

## Antibodies

For immunoblots: Anti-CypA (BML-SA296-0100, Enzo, 1:2000). Anti-CypB (ab16045, Abcam, 1:1400). Anti-β-actin (clone AC-15, ab6276, Abcam, 1:10000). Anti-VHL (sc135657, Santa Cruz Biotechnology, 1:100). IRDye® 680LT goat anti-mouse (926-68020, LI-COR Biosciences, 1:15000). IRDye® 680LT goat anti-rabbit (926-68021, LI-COR Biosciences, 1:15000). IRDye® 800CW goat anti-mouse (926-32210, LI-COR Biosciences, 1:10000). IRDye® 800CW goat anti-rabbit (926-32211, LI-COR Biosciences, 1:10000).

For flow cytometry: Anti-CypA (clone 1F4-1B5, ab58144, Abcam, 1:1000). Alexa Fluor® 488 goat anti-mouse (405319, BioLegend, 1:400). Alexa Fluor® 647 goat anti-mouse (405322, BioLegend, 1:400). FITC anti-HIV-1 Gag (KC57-FITC, clone FH190-1-1, 6604665, Beckman Coulter, 1:100). APC anti-CD3 (SK7, 981012, BioLegend, 1:100). PE anti-CD4 (SK3, 980804, BioLegend, 1:100).

## Cell lines

Calu-3 (ATCC HTB-55), HEK293T (ATCC CRL-3216), Huh7 (kindly provided by Joe Grove), and U87 CCR5 expressing cells (ATCC HTB-14) were maintained in DMEM (Gibco) with 10% foetal bovine serum (FBS) (LabTech), 100 U/mL penicillin and 100 μg/mL streptomycin (Pen/Strep; Gibco). Jurkat T cells (Clone E6-1; ATCC TIB-152) were cultured in RPMI (Gibco), 10% FBS and Pen/Strep (Gibco). THP-1 Dual reporter cells (Invivogen) were cultured in RPMI, 10% FBS, Pen/Strep, 25 mM HEPES (Sigma), 10 μg/mL of blasticidin (Invivogen) and 100 μg/mL of Zeocin™ (Invivogen).

## Isolation of primary MDMs and CD4+ T cells from peripheral blood

Peripheral blood mononuclear cells (PBMCs) were isolated from buffy coats or leukocyte cones from healthy donors (UK NHS Blood and Transplant Service) or from fresh blood from healthy volunteers by density gradient centrifugation using Lymphoprep (Stemcell Technologies). The study was approved by the joint UCL/UCLH NHS Trust Human Research Ethics Committee, and written informed consent obtained from all participants. For isolation of monocyte-derived macrophages (MDMs), PBMCs were washed three times with PBS and plated to select adherent cells. Non-adherent cells were washed away after 2 h and the remaining cells were incubated in RPMI (Gibco) supplemented with 5% heat-inactivated pooled human serum (H3667, Sigma Aldrich) and 100 ng/mL macrophage colony-stimulating factor (Peprotech). The medium was replaced after 2 days with RPMI with 5% human serum, removing any remaining non-adherent cells. Cells were treated with Cyp inhibitors 3 days later.

Resting CD4+ T cells were isolated from total PBMCs by negative selection using the human CD4+ T cell Isolation Kit (130-096-533, Miltenyi Biotec). Following isolation, cell purity was assessed by cell surface staining with CD3-APC (981012, BioLegend) and CD4-PE (980804, BioLegend) (performed in PBS), followed by analysis by flow cytometry (≥ 95% CD3+ CD4+ for all donors) (Supplementary Fig. 15a). T cells were cultured in RPMI with 10% FBS and 10 IU/mL IL-2 (Centre For AIDS Reagents (CFAR), National Institute of Biological Standards and Control (NIBSC), UK). Where indicated, T cells were activated for 4 days in culture medium in the presence of 1 μg/mL plate-bound anti-CD3 antibody (clone OKT3, 16-0037-85, Thermo Fisher Scientific) and 2 μg/mL soluble anti-CD28 antibody (clone CD28.2, 16-0289-85, Thermo Fisher Scientific).

## Screening Cyp inhibitors

U87 cells ($5 \times 10^4$ cells/well, 12 well plate), Huh7 cells ($1 \times 10^5$ cells/well, 12 well plate), Calu-3 cells ($1 \times 10^5$ cells/well, 12 well plate), Jurkat cells ($5 \times 10^5$ cells/mL, 24 well plate, 1 mL/well), THP-1 Dual cells ($4 \times 10^5$ cells/mL, 24 well plate, 1 mL/well), primary MDMs (confluent well), or primary resting/activated T cells ($1-2 \times 10^6$ cells/well, 96 well plate) were treated with 5 μM Cyp-PROTAC or DMSO for 48 h. To determine if degradation was PROTAC-mediated, Jurkat cells were pretreated for 6 h with 1 μM NEDD8-activating enzyme inhibitor MLN4924 (951950-33-7, Sigma Aldrich), or pretreated for 2 h with 50 μM VHL inhibitor (VHLi) VH298 (SML1896, Sigma Aldrich), prior to treatment with 5 μM Cyp-PROTAC or DMSO for 48 h. MLN4924 was washed out before the addition of Cyp-PROTAC, but VHLi was kept in for the duration of the experiment. In competitor experiments, Jurkat cells were treated with 10 μM TWH106 and 1 μM Cyp-PROTAC simultaneously for 48 h. For the time course experiments, Jurkat cells were treated with 5 μM Cyp inhibitor or DMSO and cells were harvested at indicated time points. For the titration experiments, Jurkat cells were treated with 0.01 – 10 μM Cyp inhibitor or DMSO for 24 h or 48 h. Before harvesting for immunoblot, cells were washed twice in PBS.

## Immunoblot

Cells were lysed in RIPA buffer (50 mM Tris pH 8.0, 150 mM sodium chloride, 1.0% Triton X-100, 0.5% sodium deoxycholate, 0.1% sodium dodecyl sulphate (SDS)) supplemented with protease inhibitors (cOmplete mini, EDTA-free, Roche) and phosphatase inhibitors (PhosSTOP, Roche). After 10 min incubation on ice, samples were centrifuged ($16000 \times g$, 10 min, 4 °C), and the supernatant protein content was determined using Pierce™ BCA Protein Assay Kit (Thermo Fisher Scientific). Samples were combined with 4x Laemmli sample buffer (200 mM Tris pH 6.8, 40% glycerol, 8% SDS, 0.4% Bromophenol Blue) and the appropriate amount of RIPA buffer to normalise protein concentrations of samples and boiled for 5 min. Proteins were separated by SDS-PAGE on 15% polyacrylamide gels

and transferred onto nitrocellulose membrane using the Trans-Blot Turbo Transfer System (Bio-Rad). Blots were blocked in 5% milk in PBS-Tween for 1 h and incubated with primary antibodies in 5% milk in PBS-Tween overnight. After washing in PBS-Tween, blots were incubated with IRDye® fluorescent secondary antibodies (LI-COR) for 1 h, washed in PBS-Tween, then PBS, and visualised on an LI-COR Odyssey imaging system. Uncropped and unprocessed scans are provided in the Source Data file (main figures) and in the Figshare research data repository https://doi.org/10.6084/m9.figshare.c. 7387381 (supplementary figures). For some experiments, samples deriving from the same experiment were run twice and blots processed in parallel (for example, to allow blotting of both CypA and CypB in parallel). Protein markers are indicated in kDa throughout. Densitometry analysis was performed using ImageJ (v2.14.0), and normalised protein densities were calculated by normalisation to β-actin loading control and untreated control as stated. Data are presented as the mean of two independent experiments.

## Proteomics

THP-1 Dual cells were treated with 5 μM Cyp inhibitor or DMSO for 72 h. After washing with PBS, cells were lysed (5% SDS, 5 mM tris(2-carboxyethyl)phosphine (TCEP), 10 mM chloroacetamide (CAA), 100 mM Tris, 1% Triton X-100, pH 8.5) and boiled for 10 min followed by sonication in a water bath for 10 min. Protein concentration was estimated by BCA assay (Thermo Fisher Scientific). Protein digestion was automated on a KingFisher APEX robot (Thermo Fisher Scientific) in a 96-well format using a protocol from Koenig et al.[83] with minor modifications. The 96-well comb is stored in plate #1, the sample in plate #2 in a final concentration of 70% acetonitrile (ACN) and with magnetic MagReSyn Hydroxyl beads (ReSyn Biosciences) in a protein/bead ratio of 1:2. Washing solutions are in plates #3–5 (95% ACN) and plates #6–7 (70% ethanol). Plate #8 contains 300 μL digestion solution of 100 mM Tris pH 8.5 and trypsin (Promega) in an enzyme:protein ratio of 1:100. The protein aggregation was carried out in two steps of 1 min mixing at medium mixing speed, each followed by a 10 min pause. The sequential washes were performed in 2.5 min at slow speed, without releasing the beads from the magnet. The digestion was set to 16 h at 37 °C with slow speed. Protease activity was quenched by acidification with trifluoroacetic acid (TFA) to a final pH of 2, and the resulting peptide mixture was purified on the OASIS HLB 96 well plate (Waters). Peptides were eluted twice with 100 μL of 50% ACN and dried in a Savant DNA120 (Thermo Fisher Scientific).

Peptides were then dissolved in 0.5% TFA before liquid chromatography–tandem mass spectrometry (MS/MS) analysis. The mixture of tryptic peptides was analysed using an Ultimate3000 high-performance liquid chromatography system coupled online to an Eclipse mass spectrometer (Thermo Fisher Scientific). Buffer A consisted of water acidified with 0.1% formic acid, while buffer B was 80% ACN and 20% water with 0.1% formic acid. The peptides were first trapped for 1 min at 30 μL/min with 100% buffer A on a trap (0.3 mm by 5 mm with PepMap C18, 5 μm, 100 Å; Thermo Fisher Scientific); after trapping, the peptides were separated by a 50 cm analytical column (Acclaim PepMap, 3 μm; Thermo Fisher Scientific). The gradient was 7 to 35% B in 103 min at 300 nL/min. Buffer B was then raised to 55% in 3 min and increased to 99% for the cleaning step. Peptides were ionised using a spray voltage of 2.1 kV and a capillary heated at 280 °C. The mass spectrometer was set to acquire full-scan MS spectra (350–1400 mass/charge ratio) for a maximum injection time set to Auto at a mass resolution of 60,000 and an automated gain control (AGC) target value of 100%. For MSMS fragmentation we chose the DIA approach: AGC target value for fragment spectra was set at 200%. 60 windows of 10 Da were used with an overlap of 1 Da (m/z range from 380 to 980). The resolution was set to 15,000 and IT to 40 ms. The normalised collision energy was set at 30%.

All raw files were analysed by DIA-NN v1.8.1[84], searching against library generated automatically using Human reference proteome (UniProt) and standard settings: peptides from 7 to 30 amino acids, max number of missed cleavages of 1, oxidation (M) and protein-acetylation as only variable modifications. The data files generated by DIA-NN were analysed using Perseus version 2.0.10.0[85]. The $\log_2(x)$ intensities were normalised by subtracting the median intensities of each replicate across all samples, followed by the median intensities of each protein within replicate groups. Two-sided two-sample t-tests were performed to identify proteins with statistically significant changes between conditions using stringency parameters s0 = 0.5 and FDR = 0.05.

## Production of lentiviral vector in HEK293T cells

HEK293T cells were transfected with 1 μg p8.91 + 1 μg pMDG + 1.5 μg pCSGW (HIV-1 GFP) or 2.5 μg pLAI ΔEnv + 1 μg pMDG (HIV-1 LAI ΔEnv GFP) in 10 cm dishes using 10 μL Fugene 6 transfection reagent (E2692, Promega) according to the manufacturer's instructions. The transfection mix was incubated for 15 min at room temperature before adding dropwise to HEK293T cells. Media was replaced after 24 h, and supernatants containing lentiviral vector were collected at 48 h and 72 h, filtered (0.45 μm), combined and used at a multiplicity of infection (MOI) of 0.25 on U87 or Jurkat cells.

## Lentiviral infection of U87 and Jurkat cells

U87 cells ($1 \times 10^4$ cells/well, 48 well plate) or Jurkat cells ($5 \times 10^3$ cells/well, 96 well U-bottom plate) were treated with serial dilutions of Cyp inhibitors as indicated. After 48 h, the media was aspirated and replaced with fresh Cyp inhibitor dilutions and the lentiviral vector (MOI: 0.25) for a further 48 h. Experiments were carried out in triplicate.

## Production of HIV-1 NL4.3

The HIV-1 clone pNL4.3 was obtained from the CFAR, NIBSC (cat# 2006). NL4.3 stocks were produced by transfection of HEK293T cells in 150 cm$^2$ flasks with 9 μg pNL4.3 using 60 μL Fugene 6 (E2692, Promega) according to the manufacturer's instructions. Supernatants were harvested at 48 and 72 h, filtered, DNase treated, purified and concentrated by ultracentrifugation through a 20% sucrose cushion and resuspended in RPMI with 10% FBS. Viral titres were determined by measuring reverse transcriptase (RT) activity in mU by SG-PERT assay[86].

## Wild-type HIV-1 spreading infection in activated primary CD4+ T cells

To infect activated primary CD4+ T cells with HIV-1, cells were mixed and incubated with HIV-1 NL4.3 virus (2000 mU RT/10$^6$ cells) for 4 h at 37 °C. This is effectively a low MOI, and the percentage of infection (% Gag+ cells) varied between 15% and 35% at 4 days post infection (dpi), depending on the donor. After exposure to the virus, T cells were washed twice in PBS by centrifugation (400 × g, 5 min) and resuspended in fresh culture medium. At the indicated time points, cells were harvested to determine intracellular infection levels by Gag staining and intracellular protein levels by CypA staining, and/or culture supernatant was harvested to measure virus release by SG-PERT[86]. Cells were supplemented with fresh IL-2 at 4 dpi. Pretreatment of T cells was with 1 μM or 5 μM Cyp inhibitor or DMSO as described. Treatment was maintained throughout the experiment or Cyp inhibitors were washed out prior to HIV-1 infection as indicated. To demonstrate PROTAC-mediated antiviral activity, cells were treated for 2 h with 50 μM VHL inhibitor (VHLi) VH298 (SML1896, Sigma Aldrich) before addition of 5 μM Cyp inhibitor for 48 h prior to washing cells and infection with HIV-1 as described above. For this experiment, Cyp inhibitors were washed out prior to infection, but VHLi treatment

was maintained for the duration of the experiment. Experiments were carried out in duplicate for each donor.

## Production of HCV subgenomic replicon RNA
HCV JFH-1 subgenomic replicon (SGR) (pFKI389Luc/NS3-3'_dg JFH, previously described)[87] plasmid DNA (10 µg) was linearised by digestion with MluI, and purified linearised DNA (1 µg) was used as a template for in vitro transcription, according to the MEGAscript T7 Transcription Kit instructions (Invitrogen). RNA was resuspended in nuclease-free water at a concentration of 1 µg/µL, aliquoted, and stored at − 80 °C.

## Electroporation of HCV replicon in Huh7 cells
HCV SGR RNA (5 µg or 1 µg) was electroporated into $2 \times 10^6$ or $2 \times 10^5$ Huh7 cells, respectively, using a Neon transfection system (Thermo Scientific). Single-cell suspensions were washed with PBS and resuspended in 100 µL or 10 µL of Buffer R, respectively. Resuspended cells were mixed with RNA and loaded into a Neon Tip and electroporated using the Neon Transfection system (1400 V, 20 ms, 1 pulse) and resuspended in 12 mL or 1.2 mL of media, respectively, prior to seeding in 96 well plates at a density of ~$2 \times 10^4$ cells/well. Pretreatment of cells was with 1 µM or 5 µM Cyp inhibitor or DMSO as described. Treatment was maintained throughout the experiment or Cyp inhibitors were washed out prior to HIV-1 infection as indicated. To demonstrate PROTAC-mediated antiviral activity, cells were treated for 2 h with 50 µM VHL inhibitor (VHLi) VH298 before the addition of 5 µM Cyp inhibitor for 48 h prior to washing cells and electroporation with HCV SGR RNA as described above. For this experiment, Cyp inhibitors were washed out prior to infection, but VHLi treatment was maintained for the duration of the experiment. Experiments were carried out in triplicate. Firefly luciferase activity was measured on a Glomax (Promega) at 4 h post electroporation (hpe) and 48 hpe using the SteadyGlo reagent (Promega) according to the manufacturer's instructions. Data at 48 hpe were normalised to the input luciferase signal at 4 hpe.

## Viability assay
3-(4,5-Dimethyl-2-thiazolyl)-2,5-diphenyl-2H-tetrazolium bromide (MTT) assay was performed on U87 cells, Jurkat cells or Huh7 cells treated with Cyp inhibitors as indicated. At the time point indicated, 10% v/v MTT (stock at 5 mg/mL in PBS) (475989, Sigma Aldrich) was added, and cells were incubated for 1 - 2 h at 37 °C. 100 µL solubilisation solution (10% SDS and 0.01 M HCl) was added, and after overnight incubation at 37 °C, absorbance was measured at 570 nm. Experiments were carried out in triplicate.

## Flow cytometry
For HIV-1 GFP lentiviral infection of U87 and Jurkat cells, cells were fixed in 4% formaldehyde in PBS. The % GFP+ (infected) cells were determined using a NovoSampler Pro (Agilent) flow cytometer. For primary T cell infection experiments, cells were washed in PBS and stained with fixable Zombie R685 Live/Dead dye (423119, BioLegend, 1:500) for 5 min at 37 °C. Excess stain was quenched with FBS-complemented RPMI and cells were fixed with 4% formaldehyde before intracellular staining. Permeabilisation for intracellular staining of primary T cells and Jurkat cells was performed with Intracellular Staining Perm Wash Buffer (421002, BioLegend) according to the manufacturer's instructions. Where appropriate, intracellular staining for HIV-1 Gag (clone FH190-1-1, Beckman Coulter, 1:100) was performed for 30 min at room temperature. Intracellular CypA was detected by incubating permeabilised cells with CypA antibody (clone 1F4-1B5, ab58144, Abcam, 1:1000) for 30 min, followed by washing (PBS) and 15 min incubation with secondary anti-mouse Alexa-Fluor488- or AlexaFluor647-tagged antibody. After washing with PBS ($700 \times g$, 5 min, 4 °C) to remove excess antibodies, data were acquired on a NovoSampler Pro (Agilent) and analysed using FlowJo v10.10 (Tree Star) or NovoExpress 1.5.0 (Agilent). Gating strategies are shown in Supplementary Fig. 15.

## Statistics
Statistical analyses were performed using GraphPad Prism (v9.5.1) as indicated in the figure captions. $DC_{50}$s and $D_{max}$s were calculated from sigmoidal (4-parameter logistic) fits, mean ($n = 2$ independent experiments).

## Bioinformatic analyses and molecular modelling
Sequence alignments were performed using the Clustal Omega web tool (https://www.ebi.ac.uk/jdispatcher/msa/clustalo) and sequence identities for the cyclophilin domains were obtained from BlastP (https://blast.ncbi.nlm.nih.gov/Blast.cgi). All rendering of protein structures, surface generation and similarity colouring was performed using the MOE software (2020.0901). For surface similarity, the CypA structure (1YND) was used as a reference and amino acids of other Cyps were coloured according to their similarity to CypA using the Blosum62 scoring matrix embedded in MOE. Molecular docking of TWH106 in CypA (1YND) was performed with the MOE software in the Amber10:EHT field using the built-in induced fit docking module. A comparison of the CsA (1CWA) and TWH106 binding modes were performed by protein alignment in the MOE software.

## Reporting summary
Further information on research design is available in the Nature Portfolio Reporting Summary linked to this article.

## Data availability
All raw data for the main figures are provided in the Source Data file provided with this paper. Raw data for the supplementary figures is available in the Figshare research data repository https://doi.org/10.6084/m9.figshare.c.7387381. Characterisation data for synthesised compounds, including LC traces, NMR and MS are provided in the Supplementary Information file. The associated raw data and processed NMRs are available in the Figshare research data repository https://doi.org/10.6084/m9.figshare.c.7387381. The mass spectrometry proteomics data have been deposited to the ProteomeXchange Consortium via the PRIDE[88] partner repository with the dataset identifier PXD057024. Molecular modelling studies were performed using the following accession codes: 1YND, 1CWA. Supporting files for molecular modelling and are available in the Figshare research data repository https://doi.org/10.6084/m9.figshare.c.7387381.

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

## Acknowledgements

The results presented here formed part of the PhD thesis of Lydia S. Newton[57]. The authors acknowledge the support of FastForward, National Multiple Sclerosis Society grants G551-6235 and FF-1502-03239 (D.L.S.), Wellcome Investigator Awards (220863, G.J.T. and 223065, C.J.), a Wellcome Collaborative award (214344, G.J.T., D.L.S.), Wellcome Trust Rapid Response call 204841/Z/16/Z, The Rosetrees Trust (ID2020/100020, D.A., G.J.T., D.L.S.), an MRC award (MR/S023380/1, S.R., G.J.T.), a Wellcome Trust Multiuser Equipment grant (221521/Z/20/Z) (K.T.), the UCL-Birkbeck MRC doctoral training partnership (MR/N013867/1, L.S.N.) and the Moorfields Eye Hospital NHS Trust (C.G.). The work of Ciulli laboratory on VHL-based PROTACs has received funding from the European Research Council (ERC) under the European Union's Seventh Framework Programme (FP7/2007-2013) as a starting grant to A.C. (grant agreement ERC-2012-StG-311460 DrugE3CRLs) and the Innovative Medicines Initiative 2 (IMI2) Joint Undertaking under grant agreement no. 875510 (EUbOPEN project to A.C.) from the European Union's Horizon 2020 Research and Innovation Programme.

## Author contributions

L.S.N., C.G., S.R., R.J.S., G.J.T. and D.L.S. conceived the study. L.S.N., C.G., S.R., R.J.S., A.J.W., T.W.H., K.L.M., D.A., R.Z.C., A.K.R., M.L.G., Y.Y.T.,

L.G.T., C.J., K.T., A.C., G.J.T. and D.L.S. designed the experiments and interpreted the data. L.S.N., C.G., S.R., R.J.S., T.W.H., A.J.W., K.L.M., D.A. and R.Z.C. performed the experiments. A.C. supervised the surface plasmon resonance, size exclusion chromatography and fluorescence polarisation experiments designed and performed by A.J.W. K.T. and G.J.T. supervised the proteomics experiments designed and performed by R.Z.C., L.S.N., K.L.M. and D.A. G.J.T. supervised the cell biology and virology experiments designed and performed by L.S.N. and S.R. C.J., G.J.T. and A.K.R. supervised the HIV-1 NL4.3 infection experiments designed and performed by L.S.N. D.L.S. supervised the chemical synthesis designed and performed by C.G., R.J.S., T.W.H. and K.L.M. and the surface plasmon resonance experiments and structural modelling designed and performed by C.G. L.S.N., C.G., G.J.T. and D.L.S. wrote the manuscript with input from all other authors.

## Competing interests

The Ciulli laboratory receives or has received sponsored research support from Almirall, Amgen, Amphista Therapeutics, Boehringer Ingelheim, Eisai, Merck KGaA, Nurix Therapeutics, Ono Pharmaceutical and Tocris-BioTechne. A.C. is a scientific founder, advisor, and shareholder of Amphista Therapeutics, a company that is developing targeted protein degradation therapeutic platforms. The other authors declare no competing interests.
