## [Transparent Peer Review file · Nature Communications]

Macrocycle-based PROTACs selectively degrade cyclophilin A and inhibit HIV-1 and HCV

Corresponding Author: Professor David Selwood

Version 0:

Reviewer comments:

Reviewer #1

(Remarks to the Author)

This is an excellent article describing the synthesis of PROTACs derived from SfA that specifically target CypA leading to its degradation via the ubiquitin pathway in primary cells as well as cell lines.

Interestingly, these synthesized PROTACs degrade CypA more efficiently than CypB although both Cyps bind tightly to SfA.

Further demonstrating the CypA requirement for efficient HIV-1 replication in human cells, the addition of synthesized PROTACs to target cells significantly reduce viral infection.

Reviewer #2

(Remarks to the Author)

This is a well-written manuscript describing the development of proteolysis targeting chimeras (PROTACs) of cyclophilin A (CypA) and their application to inhibit HIV-1 replication in cell culture. For this work, the authors undertook a significant synthetic chemistry effort to develop a simplified cyclophilin ligand based off of a previously described synthetic macrocycle, Sanglifehrin A (SfA). A rational effort to engineer specificity for CypA over CypB was attempted based on computational docking, leading to synthesis of TWH106. The authors generated two PROTACs based on attachment of a linker and VHL ligand off of the solvent-exposed mTyr to produce CG167 or off of the macrocycle ester to produce RJS308. The authors then performed characterization of these compounds in experiments to evaluate their activity in

- an assay for neddylation-dependent CypA degradation (Figure 3)
- an assay examining CypA versus CypB depletion in a panel of cell lines and both resting and activated primary T cells (Figure 4)
- washout experiments in which primary resting CD4+ T cells were treated for 48 hours, and then monitored for an additional 72 hours +/- PROTAC (Figure 5)
- antiviral assays conducted in cell lines and primary activated CD4+ T cells (Figure 6)

The principal conclusions from this work drawn by the authors are (1) CG167 and RJS308 induce degradation of CypA by a PROTAC mechanism; (2) degradation is selective for CypA in cell lines and primary cells, which contrasts with the inhibitory activity of parental compound TWH106 and the activity of a previously published CsA-based PROTAC, JW4-10; (3) degradation of CypA by these PROTACs is associated with antiviral activity against HIV-1; (4) the CypA PROTACs have superior antiviral activity compared to the parental inhibitor

These conclusions are exciting and are anticipated to attract broad interest from both the antivirals and targeted protein degradation communities. The work is multidisciplinary and nicely combines an impressive chemistry effort with cell biology and virology experiments. While I am very enthusiastic about the work, there are some notable shortcomings in the characterization of the activity and selectivity of the compounds and the extent to which antiviral activity of CG167 and RJS308 derives from the PROTAC mechanism versus the CypA-inhibitory activity of the TWH106 moiety. I have written about this in detail below.

Major:

1. Is antiviral activity due to the PROTAC mechanism and is the PROTAC mechanism responsible for the "improved antiviral

activity against primary T cells compared to the non-PROTAC parental inhibitor.” While the experiments show that CG167 and RJS308 both inhibit HIV-1 and induce depletion of CypA, they do not address the possibility that the PROTACs may also act as CypA inhibitors and that antiviral activity may also come from inhibition of CypA, in much the same way that THW106 has antiviral activity. This is a major concern.

- a. The authors should show to what extent antiviral activity is lost when the PROTAC mechanism is blocked. Ideally, this would be with negative compounds in which the VHL-binding moieties of CG167 and RJS308 are modified to be the stereoisomers that have significantly lost interaction with the VHL E3 ligase.
- b. The concentration of PROTAC used and the duration of compound treatment should be chosen to minimize the potential for off-target and off-mechanism effects.

2. Concentration dependence of CypA degradation by CG167 and RJS308

- a. In most cases, the authors have drawn conclusion regarding selectivity and dose-responsiveness based on at most two concentrations (1 and 5 μM). The rationale for choosing 5 μM for most of the experiments is not provided, but should be. In experiments examining dose-response for 1 versus 5 μM , it's not clear where on the titration curve this is – it's possible that the lack of dose-response in some experiments (e.g., CG167 in Huh7 cells, Figure 4) that 1 and 5 μM are already greater than the DC_{max} so additional degradation/depletion is not observed. The authors should report DC_{50} and DC_{max} (standard metrics in the TPD field) against CypA and CypB under experimental conditions that are relevant for interpreting their antiviral activity experiments so that readers can evaluate whether or not degradation of CypA by the PROTACs is correlated with antiviral activity.
- b. Why is the extent of depletion of CypA in Jurkat cells significantly different in Figure 4e-f versus Figure 6d? There is considerably less depletion in Figure 6d despite these cells having been treated for longer (96 hours in Fig 6D versus 48 hours Figure 4e-f). If anything, one would assume greater depletion with longer treatment time. That there is less depletion at 96h raises concerns that observations made at these long time points reflect more than just degradation of CypA by the PROTAC.

3. Time-dependence of CypA degradation by CG167 and RJS308

a. Time-dependence of PROTAC activity

Experiments showing the time-dependence for target degradation is also something fairly standard in the TPD field but noticeably absent in the manuscript. The authors generally monitored CypA and CypB abundance after 48-72h of treatment with the PROTAC. This is a long duration and requires some explanation.

b. Prolonged PROTAC treatment and conclusions regarding PROTAC mechanism and specificity

If 48-72h of treatment are required to observe significant depletion of CypA, then this raises some concern that the depletion may not be due to PROTAC-directed degradation of CypA but instead affected by an indirect mechanism (or mechanisms) affecting CypA expression or turnover. This concern seems warranted given the evidence for some CypB depletion after longer treatments (Figure 4, Supplementary Fig 8). Is CypB actually being degraded by the PROTAC mechanism or are there other mechanisms at play? The authors should provide data to distinguish between these two possibilities and/or modify their text to acknowledge that they cannot distinguish between these two sets of possibilities using only the current data.

c. Interpretation of washout experiments and conclusion for prolonged-effect of CypA PROTACs

Are the authors certain that the washout experiments worked? Once the ternary complex has formed and the target has been ubiquitinated, degradation of the target protein is usually fairly rapid. Figure 5's data showing that CypA degradation has not reached the maximum at 48 hours treatment and that the extent of CypA depletion is even greater following another 72 hours (120 hours total) is a puzzling result, in particular that the extent of depletion increases for 72h after the washout (Figure 5 condition 2). As noted above, the prolonged treatment times raise concerns that other mechanisms affecting CypA abundance are involved but are not being acknowledged. I also don't think that the authors' explanation that additional CypA is being degraded by the PROTAC mechanism even when the PROTAC was washed out makes sense. The prolonged pharmacological effect of PROTACs is generally due to the requirement for new protein synthesis to restore the function, which contrasts with the rapid regain of function upon dissociation of inhibitors. In Figure 5, it seems more likely that the washout was not effective. Therefore, I don't think that the claim that PROTAC-driven loss of CypA is long-lasting (lines 207-208) is solid without better evidence that their washout worked.

4. Selectivity of CG167 and RJS308

a. Global selectivity

The authors address selectivity by comparing degradation of CypA and CypB; however, this analysis is inadequate to support the statement that CG167 and RJS308 are selective for CypA. It is well-established that PROTACs can induce degradation of neosubstrates that may not be predicted. As a consequence, the targeted protein degradation field commonly utilizes (and requires) proteomic profiling experiments, ideally performed with the PROTAC as well as with a negative control compound whose E3 ligand has been modified to prevent interaction with the E3 ligase. This analysis is of particular concern given the number of related cyclophilin proteins in the host cell. As recognized by the authors, the affinity of the interaction between the PROTAC and the target protein is not a predictor of the efficacy of target degradation. If the authors do not have in-house access to proteomic analysis, this can be accessed through collaboration and community efforts to define the degradable proteome.

b. Selectivity for CypA versus CypB

Although THW106 was designed to be selective for CypA over CypB, SPR experiments with recombinant proteins indicate that this is not the case. While the authors mention that the lack of difference in the SPR experiments may be due to the use of a commercially available, truncated form of CypB, it's unclear whether this is the case. This is an important issue because the selectivity of the PROTACs for CypA over CypB may be due to a difference in affinity for these two targets and/or may be due to differences in ternary complex formation. Is the superiority of the PROTAC over the parental inhibitor due to the

rational change that was made to the ligand or is it due to differences in ternary complex formation and hence due to the PROTAC mechanism? The authors should try to address this experimentally – perhaps, CETSA experiments monitoring engagement of CypA versus CypB in cellulo would be informative? They should at least discuss these issues in the context of the PROTACs' selectivity for CypA versus CypB when compared to the lack of selectivity for TWH106 and JW4-10. The explanation provided in lines 192-196 is inadequate given what is known about determinants of PROTAC selectivity.

5. Data in Figure 4i and 4j are not discussed in the text.

a. What are the conclusions from these figures?

b. Are the graphs in 4j derived from the blots in 4i or are the data in 4i from one donor and the data from 4j from another donor?

c. Why do CG167 and RJS308 cause an increase in CypB in Figure 4j? Potential mechanisms should be discussed. Also why isn't the increase in CypB apparent in the 4j plots apparent in Figure 4i? I think that the plots combine data in Figure 4i with data from Supplementary Figure 7, but this should be explained in the Figure legend to make it clearer for readers.

Minor:

1. a. In Figure 3, a control is needed showing the effect of MLN4924 along on CypA is needed. MLN4924 and proteasome inhibitors can lead to increased steady-state abundance of proteins. In this case, the neddylation inhibitor does not necessarily "rescue" the target protein from the PROTAC mechanism.

Reviewer #3

(Remarks to the Author)

This article by Gathmann et al. presents PROTACs aimed at Cyclophilin A (Cyp A), a protein cofactor exploited by viruses during infection. In this paper, the authors describe a PROTAC based on a new peptide macrocycle derived from Sanglifehrin A. This paper builds upon some of the authors previous work targeting CypA, this new heterobifunctional molecule is more specific for CypA over CypB and is another good example of the use of PROTAC as an antiviral treatment, and as the recent Covid19 pandemic has highlighted, new small molecules are needed. The authors demonstrated that at least one of their molecules acts through a PROTAC mechanism and was able to inhibit HIV-1 replication in human T cells. The manuscript is well written, and the figures are particularly clear and readable. This manuscript should appeal to the readers of Nature Communications and should be published after some revisions and clarifications from the authors.

1) Line 100: The authors mention SAR studies to develop the inhibitor TWH106. Was the SAR fully based on in silico docking or was it based on SPR binding as done with the PROTACs and the inhibitor as described later? It may be interesting to mention it in the main text.

2) Line 118-119: The authors focus their PROTAC synthesis on very specific linkers, namely linkers of 10 or 12 atoms with a triazole stemming from the cycloaddition of the two halves of the molecule. In heterobifunctional molecules, linker length and composition play a major role in degradation efficiency. In their previous report (Colpitts et al. eLife 2002) of a CypA PROTAC using cyclosporin A as a warhead, the linker used is of only 3 atoms. How did the authors choose this specific linker length and were several other linkers (length or composition) tested? It may be worth mentioning the author's efforts to develop the most potent PROTAC possible. On the same topic, were other E3 ligases considered during the PROTAC development, namely using thalidomide derivatives or other attachment points on the VHL ligand?

3) Figure 3 and CypA degradation by a PROTAC mechanism:

a. The authors demonstrate that their molecules act by PROTAC mechanism by using a NEDD8ylation inhibitor and by competition with free inhibitor. In these degradation blots, each data point has been tested as a single technical replicate and as a biological duplicate. In some cases, Figure 3a and 3b for CG167 and Figure 3b for RJS308, the degradation levels vary from 50% to close to 100%. A third biological replicate would be beneficial to conclude on the degradation potential of these molecules, since the two data points are so different.

b. It looks like in the experiment 2, the degradation levels observed are lower across all conditions tested, could this experiment be an outlier?

c. The authors mention that a ternary complex between CG167, CypA and VCB could not be observed. Moreover, in supplementary Figure 5b: Was the competition FP experiment not run with CG167 or was no competition observed? The lack of binding of CG167 to VHL would suggest that this compound may be able to degrade CypA through a non-PROTAC mechanism. Has the ability of this PROTAC to bind VHL been verified by the authors?

4) CypA degradation by a PROTAC mechanism and Degradation of cyclophilins A and B (Figure 3 and Figure 4):

a. The authors do not show any clear dose response or time course of their PROTAC activity. How were the treatment conditions selected?

b. In their previous paper (Colpitts et al. eLife 2002), the authors showed that their previous PROTAC was able to degrade CypA, as low as 30 nM and that 48h treatment was only required to reach 100% degradation. Have the authors compared the two new PROTACs to their previous molecule in a time course or in a wider dose response?

c. The SPR experiment shows about a 100 nM affinity of the PROTACs for their target, is 1 μM treatment needed or is this due to poor cell permeability of the molecules?

5) Degradation of cyclophilins A and B (Figure 3 and Figure 4):

a. When comparing the degradation selectivity of their PROTACs, the authors show that JW4-10, a cyclosporin A based

PROTAC can degrade CypB. As mentioned earlier, in the biological replicate of this experiment, degradation of CypB was observed to a much lower level. A third data point would be welcome to validate the extent of CypB degradation.

b. The authors mention the weaker degradation potential of their PROTACs in U87 cells, but they never provide any explanation as to why this is observed. Can the authors comment on this observation?

6) Figure 6: Why was the time of pre-incubation between U87/Jurkat cells and donor T-cell changed from respectively 48 hours to 72 hours?

7) Supplementary Figure 2: SPR experiment. In this SPR experiment, the theoretical maximum RU expected is close to 220 RU. Can the authors comment on why such a high density of protein was needed? Despite having a similar size, the two PROTAC show very different binding behaviors, namely RJS308 binds with a maximum RU of 130, while CG167 only binds with a 65 RU maximum. Do these PROTAC bind in different pockets, or do they bind a different subset of the protein population immobilized on the surface? What does this weaker RU entail for CG167?

8) Supplemental information: The addition of the NMR spectrum for each small molecule synthesized on top of the peak report would be appreciated for full transparency. However, since the authors did a thorough analysis of their final compounds, the NMR spectra may not be required and there is no doubt that they did synthesize the desired molecules.

Version 1:

Reviewer comments:

Reviewer #2

(Remarks to the Author)

The authors have done a tremendous job adding in new experiments to demonstrate that CG167 and RJS308 act via a TPD mechanism. I think this is a great study; however, I don't think that all of the conclusions/claims are supported by the data. I recommend acceptance if the following are addressed:

Major issues:

1. Since cyclophilin H and E came up as significant targets of both PROTACs, the authors should provide data showing whether either or both contribute to the antiviral activity of CG167 and/or RJS308. CypE has come up as a restriction factor for influenza viruses and given its nuclear localization could affect HIV. Alternatively, authors should add text acknowledging that this is a possibility and that their reported experiments do not bear on this.

2. The authors claim that the Cyp PROTACs have greater potency at lower concentrations compared to the Cyp inhibitor. I think this is true at a single concentration (1 μ M) and a single time point (day 4); however, the PROTACs have lower potency as measured by reduced viral RNA at 5 μ M on both days 4 and 8, and antiviral efficacy at the effective dose is also LOWER for the PROTACs. The antiviral activities are the same for the PROTACs and the inhibitor at the last time point in the Gag+ assay (Figure 6b-6e, 6n-6o) and the RT assay (Fig 6r-6s). The inhibitor also looks more potent and/or more efficacious in Supplementary Figures 12b-12c, 13a-13b, 14a-14c (some but not all plots). It's hard to interpret the data because only two concentrations are being reported, the abundance of CypA (or CypE and CypH) in these cells is not known, and the time points are long. A titration curve for the antiviral activity is really what's needed here along with immunoblot (or perhaps intracellular staining quantified by flow cytometry) data indicating how much CypA has been depleted. I realize it's hard to conduct these experiments, but then the conclusions need to be moderated to avoid over-interpretation of the data.

3. Likewise for the HCV antiviral data: Authors need to discuss that the inhibitor has greater antiviral efficacy at its effective dose (10 μ M) than the PROTACs. It would be better if they had a titration and could discuss the EC50 values of the PROTACs versus the inhibitor.

4. Time points: Shouldn't Cyp inhibition happen upon binding and so antiviral activity due to loss of CypA function should happen with the same or faster kinetics than antiviral activity due to targeted degradation of CypA induced by the PROTAC? Why in all of the time courses does the inhibitor's antiviral activity lag that of the PROTACs? This raises the concern that the antiviral activity is not due to loss of CypA activity. Does CypA have a scaffolding function that is not affected by the inhibitor but is affected by the PROTAC?

5. Description of the HIV-1 experiments is confusing. What was the viral challenge? Multiplicity of infection should be reported. "...cell swere mixed and incubated with 2000 mU reverse transcriptase of HIV-1 NL4.3" doesn't make sense. Did you transfect in viral RNA or did you infect?

6. Lines 73-79: It's not clear that PROTACs can address the liabilities that prevented advancement of CypA inhibitors against HCV and HIV-1. The Cyp inhibition had adequate potency, the limiting issue has been that the viruses are not as dependent on CypA activity. Also, as one gets into the data presented, TWH106 and the two PROTACs have equivalent antiviral activity at later time points in Figure 6 and antiviral efficacy (max inhibition of virus) is higher for the inhibitor. This language needs to be moderated.

Minor questions:

7. Sentence "Antiviral PROTAC development has also gained interest for its potential to reduce sensitive to resistance" should cite reference 72 as well since this is the one publication showing this can be possible.

8. Do modeling and MD simulations support the hypothesis that the exit vector is the issue for CG167?

Some comments on figures and legends – the data are very nice, but hard to take in. These suggestions are meant to make it easier for your readers:

9. Figure 4a-e: Indicate what time post-infection the samples were collected on the graph or provide schematics to help readers understand the different treatment conditions. It's written in the legend, but difficult to follow.

Figure 6:

10. It was difficult to discern the grey, green, and blue colors on my screen given the size of the plots. Suggest that the authors try to make thicker lines, use colors with greater contrast, and/or add legends for the symbols to the graph. Legend should indicate multiplicity of infection or some other metric for the size of the viral challenge in this experiment.

Version 2:

Reviewer comments:

Reviewer #2

(Remarks to the Author)

The authors' response to my questions and issues is satisfactory. I endorse acceptance of the manuscript and congratulate them on a terrific study.

Reviewer #1 (Remarks to the Author):

This is an excellent article describing the synthesis of PROTACs derived from SfA that specifically target CypA leading to its degradation via the ubiquitin pathway in primary cells as well as cell lines.

Interestingly, these synthesized PROTACs degrade CypA more efficiently than CypB although both Cyps bind tightly to SfA.

Further demonstrating the CypA requirement for efficient HIV-1 replication in human cells, the addition of synthesized PROTACs to target cells significantly reduce viral infection.

Reviewer #2 (Remarks to the Author):

This is a well-written manuscript describing the development of proteolysis targeting chimeras (PROTACs) of cyclophilin A (CypA) and their application to inhibit HIV-1 replication in cell culture. For this work, the authors undertook a significant synthetic chemistry effort to develop a simplified cyclophilin ligand based off of a previously described synthetic macrocycle, Sanglifehrin A (SfA). A rational effort to engineer specificity for CypA over CypB was attempted based on computational docking, leading to synthesis of TWH106. The authors generated two PROTACs based on attachment of a linker and VHL ligand off of the solvent-exposed mTyr to produce CG167 or off of the macrocycle ester to produce RJS308. The authors then performed characterization of these compounds in experiments to evaluate their activity in:

- *an assay for neddylation-dependent CypA degradation (Figure 3)*
- *an assay examining CypA versus CypB depletion in a panel of cell lines and both resting and activated primary T cells (Figure 4)*
- *washout experiments in which primary resting CD4+ T cells were treated for 48 hours, and then monitored for an additional 72 hours +/- PROTAC (Figure 5)*
- *antiviral assays conducted in cell lines and primary activated CD4+ T cells (Figure 6)*

The principal conclusions from this work drawn by the authors are (1) CG167 and RJS308 induce degradation of CypA by a PROTAC mechanism; (2) degradation is selective for CypA in cell lines and primary cells, which contrasts with the inhibitory activity of parental compound TWH106 and the activity of a previously published CsA-based PROTAC, JW4-10; (3) degradation of CypA by these PROTACs is associated with antiviral activity against HIV-1; (4) the CypA PROTACs have superior antiviral activity compared to the parental inhibitor

These conclusions are exciting and are anticipated to attract broad interest from both the antivirals and targeted protein degradation communities. The work is multidisciplinary and

nicely combines an impressive chemistry effort with cell biology and virology experiments. While I am very enthusiastic about the work, there are some notable shortcomings in the characterization of the activity and selectivity of the compounds and the extent to which antiviral activity of CG167 and RJS308 derives from the PROTAC mechanism versus the CypA-inhibitory activity of the TWH106 moiety. I have written about this in detail below.

Major:

1. *Is antiviral activity due to the PROTAC mechanism and is the PROTAC mechanism responsible for the "improved antiviral activity against primary T cells compared to the non-PROTAC parental inhibitor." While the experiments show that CG167 and RJS308 both inhibit HIV-1 and induce depletion of CypA, they do not address the possibility that the PROTACs may also act as CypA*

inhibitors and that antiviral activity may also come from inhibition of CypA, in much the same way that TWH106 has antiviral activity. This is a major concern.

a. The authors should show to what extent antiviral activity is lost when the PROTAC mechanism is blocked. Ideally, this would be with negative compounds in which the VHL-binding moieties of CG167 and RJS308 are modified to be the stereoenantiomers that have significantly lost interaction with the VHL E3 ligase.

This is a valid point, and the original experiments could not fully exclude that the difference in antiviral activity observed with the PROTACs compared to the inhibitors is due to having generated a better inhibitor. To address this comment, we have performed an HIV replication assay in the presence of VHL inhibitor VH298 (New Fig. 6). Co-treatment with VH298 and PROTACs fully rescued CypA levels as expected. Furthermore, the improved antiviral activity seen with the PROTACs was also abrogated and brought back to the inhibitor - TWH106 levels. Note that VH298 did not affect donor cell viability compared to other treatments at the concentrations tested. Furthermore, our PROTACs are worse binders by SPR than the original Cyp binder TWH106 (Table 1) and are expected to have poorer cell permeability due to their size and structure. When we performed the anti-HIV experiment (New Fig. 6) at a higher dose (5 μ M), TWH106 becomes indeed a better antiviral than the PROTACs, consistent with it being a better inhibitor. It is thus reasonable to assume that the improved antiviral activity seen with the PROTACs is mainly due to CypA degradation.

We are aware that epimer treatments are one way to control for this issue. However, the 14-step synthesis to be re-performed required synthetic resources we did not have when the reviews came in. We hope that the newly performed VH298 experiments compensate for this deficit.

b. The concentration of PROTAC used and the duration of compound treatment should be chosen to minimize the potential for off-target and off-mechanism effects.

In response to this comment, we have included a comprehensive characterisation of the time and concentration behaviour of our Cyp-PROTACs in the main manuscript (new Fig. 3). This addition helps rationalise our choice of treatment parameters.

1. Treatment time course: Our study employs Cyp-PROTACs as tools to investigate the influence of complete CypA removal on viral infection, requiring maximal CypA degradation. The slow action of CG167 (new Fig. 3g and see reviewer #2 point 3b) prompted us to use 48h minimum treatment time. Importantly, our data shows that the parental inhibitor TWH106 has no effect on CypA levels over a 48h treatment at 5 μ M.

2. Proteomics experiments (new Fig. 5a-c): We conducted proteomics experiments after a 72-hour treatment to assess off-target degradation for all compounds tested. The results indicated minimal off-target effects, further supporting our choice of a 48-hour treatment duration.

3. Dose response experiments: Dose-response experiments were performed at both 24h and 48h treatment times. These experiments revealed that after 48 hours maximal degradation was achieved at around 1 μ M (CG167) and 5 μ M (RJS308) (new Fig. 3n). In contrast, the parental inhibitor TWH106 exhibited a mild CypA loss only at 10 μ M and after 24 hours, and this was not observed at 48 hours (new Fig. 3n). Given the plateau observed in CypA degradation at 5 μ M for both PROTACs and the absence of off-mechanism CypA degradation for TWH106 at this concentration, we selected 5 μ M for subsequent experiments.

In summary, our choice of a 48h treatment time and 5 μ M concentration for Cyp-PROTACs is consistent with choosing the lowest effective concentration and duration required for effective degradation. The lack of off-target effects observed, despite the high abundance of CypA as a cellular protein evidences the selectivity of this approach. These considerations collectively support the rationale for our experimental design, and enable us to assess both PROTACs simultaneously, thus reducing practical expenses.

2. Concentration dependence of CypA degradation by CG167 and RJS308

a. In most cases, the authors have drawn conclusion regarding selectivity and dose-responsiveness based on at most two concentrations (1 and 5 μM). The **rationale for choosing 5 μM** for most of the experiments is not provided, but should be. In experiments examining dose-response for 1 versus 5 μM , it's not clear where on the titration curve this is – it's possible that the lack of dose-response in some experiments (e.g., CG167 in Huh7 cells, Figure 4) that 1 and 5 μM are already greater than the DC_{max} so additional degradation/depletion is not observed. The authors **should report DC_{50} and DC_{max} (standard metrics in the TPD field)** against CypA and CypB under experimental conditions that are relevant for interpreting their antiviral activity experiments so that readers can evaluate whether or not degradation of CypA by the PROTACs is correlated with antiviral activity.

We agree that determining DC_{50} and D_{max} is important for assessing dose-responsiveness. To address this concern, we have conducted full dose-response experiments in our model Jurkat cell line (new Fig. 3), and we have included the DC_{50} and D_{max} determination in our revised manuscript (lines 186-187).

The data from our experiments shows that the curve plateaus at 5 μM for RJS308, supporting our concentration choice in the manuscript. Additionally, we acknowledge that in the case of Huh7 cells, including an additional 1 μM concentration point did not yield valuable information, so have changed this to just include 5 μM concentration (new Fig. 5m). For a more comprehensive explanation regarding the concentration choice, we would like to direct the reviewer to our response in reviewer #2 point 1b.

b. Why is the extent of depletion of CypA in Jurkat cells significantly different in Figure 4e-f versus Figure 6d? There is considerably less depletion in Figure 6d despite these cells having been treated for longer (96 hours in Fig 6D versus 48 hours Figure 4e-f). If anything, one would assume greater depletion with longer treatment time. That there is less depletion at 96h raises concerns that observations made at these long time points reflect more than just degradation of CypA by the PROTAC.

Thank you for this detail-attentive comment. We attribute those variations to be due to differences between immunoblot (old Fig. 4e-f) and FACS (old Fig. 6d) quantification techniques. FACS has a smaller detection range compared to western blot, due to background cell fluorescence, even when the protein is completely depleted. For demonstration purposes, we have included here quantification of PROTAC dose-response immunoblots (left) also in new Fig. 3n, and the corresponding FACS quantification of the same cells (right), which is not

included in the paper. Regarding the concerns in kinetics, our kinetic study newly included in the main manuscript (new Fig. 3) shows that the depletion is greater with time by immunoblot, with CypA levels being lower after 4 days than 2 days.

3. Time-dependence of CypA degradation by CG167 and RJS308

a. Time-dependence of PROTAC activity

Experiments showing the time-dependence for target degradation is also something fairly standard in the TPD field but noticeably absent in the manuscript. The authors generally monitored CypA and CypB abundance after 48-72h of treatment with the PROTAC. This is a long duration and requires some explanation.

We acknowledge the importance of time-dependence experiments in the TPD field and appreciate the reviewer's suggestion. To address this, we have made the following changes to our manuscript:

Inclusion in main manuscript: Time-dependent experiments were previously found in supplementary Fig. 8, which could be easily missed. To rectify this, we have moved these experiments to the main manuscript, presenting them in new Fig. 3a-g. We also added earlier time points due to the faster kinetics of RJS308.

Choice of longer treatment times: The extended treatment times of 48-72 hours were selected due to the differences in degradation kinetics between the two PROTACs, as mentioned in reviewer #2 point 1b. CG167 has a slower degradation rate, and it required five days to achieve the same level of CypA degradation observed with RJS308 after one day (new Fig. 3g). We used 48-hour treatments as a practical compromise to examine both PROTACs simultaneously. For proteomic experiments, which are resource-intensive and time-consuming, we extended the treatment duration to ensure complete CypA removal. Note that proteomics experiment showed very clean profiles for both PROTACs despite the long treatment times.

A more detailed discussion on the slower degradation kinetics of CG167 is provided in response to reviewer #2 point 3b.

b. Prolonged PROTAC treatment and conclusions regarding PROTAC mechanism and specificity

If 48-72h of treatment are required to observe significant depletion of CypA, then this raises some concern that the depletion may not be due to PROTAC-directed degradation of CypA but instead affected by an indirect mechanism (or mechanisms) affecting CypA expression or turnover.

We appreciate the reviewer's concern and have taken steps to provide a more comprehensive understanding of the mechanisms involved. Specifically, we have focused on addressing the slow degradation kinetics observed with CG167, as RJS308 exhibited faster kinetics.

E3-dependent mechanism: We have conducted additional experiments to demonstrate the proteasome-dependent mechanism of both CG167 and RJS308. In the original manuscript, we showed that CypA is rescued from degradation by the neddylation inhibitor MLN4924 and also by an excess of the parental inhibitor TWH106. As a novel addition, we have now included an experiment showing that an excess of VH298, a cell-permeable E3 ligase inhibitor (new Fig. 4b), also rescues CypA. This additional experiment provides evidence of CG167 acting as a bifunctional PROTAC that requires binding to the VHL E3 ligase to degrade CypA.

Slow CG167-induced degradation rationale: We have provided a rationale for the slow degradation kinetics of CG167. New biochemical data by SPR, FP, and SEC (new Fig. 4f-i, new supplementary Fig. 7) indicates that RJS308 forms a ternary complex with CypA and VHL, as expected, with a low nanomolar affinity. In contrast, CG167 fails to form a ternary complex at the concentrations tested. This is likely the primary reason for its slower kinetics, as only a small fraction of CG167 engages VHL ($K_D > 2.5 \mu\text{M}$ by FP). The reasons behind this lack of engagement are unclear to this date. Sample integrity and NMR data was verified, and the corresponding inactive VHL-NH2 starting material epimer was not present in the laboratory, excluding synthesis of the wrong epimer. Our current hypothesis is that the different linker attachment may affect the binding of VHL due to steric clashes. More complex scenarios, such as a trashcan-type mechanism (degradation through CypA-altered conformation after binding by CG167, leading to proteasomal degradation, for example) is theoretically possible. The involvement of VHL as exemplified by VH298-mediated rescue makes this possibility however unlikely. We acknowledge that the lack of in vitro engagement is surprising and are working on elucidating this, as stated in line 378.

Selective action and off-target mechanisms: To address concerns about off-target effects, we emphasize that the parental inhibitor TWH106 did not affect CypA levels under the tested conditions (new Fig. 3a, d, h, k), excluding off-target effects due to the warhead. Our proteomics experiments (new Fig. 5) further confirm high selectivity of CG167 and RJS308 for CypA, making an indirect mechanism (for example, inhibition of CypA regulators) less likely.

This concern seems warranted given the evidence for some CypB depletion after longer treatments (Figure 4, Supplementary Fig 8). ***Is CypB actually being degraded by the PROTAC mechanism or are there other mechanisms at play? The authors should provide data to distinguish between these two possibilities and/or modify their text to acknowledge that they cannot distinguish between these two sets of possibilities using only the current data.***

Following this comment, we have conducted additional experiments to investigate the mechanism of CypB depletion, particularly at longer time points.

Additional time course experiments (new Fig. 5): we have included new data in our manuscript to clarify the kinetics of CypB degradation. This data indicates that there is a slight reduction in CypB levels observed at longer time points for the PROTACs, whereas THW106 results in reduction in CypB levels within 24h. THW106 alone having a greater impact on CypB levels compared to the Cyp-PROTACs supports the idea of a non-PROTAC mechanism triggered upon CypB binding solely, as suggested by reviewer #2.

To further confirm this, we performed a control experiment using our previously reported CsA-based PROTAC, JW4-10, which is known to effectively degrade CypB within 24 hours. In this case, MLN4924 could not rescue CypB to normal levels (New Supplementary Fig. 10), suggesting a neddylation-independent mechanism (i.e. not a VHL PROTAC mechanism).

Based on these findings, we have revised our manuscript to acknowledge that the observed effect on CypB is likely due to more complex mechanisms, perhaps involving CypB secretion upon inhibitor binding but which we cannot fully identify (lines 396-398).

The reviewers input led us to strengthen the scientific rigor of our study by reinforcing the proposed PROTAC mechanism for CypA and providing a more robust understanding of our PROTAC's specificity.

c. Interpretation of washout experiments and conclusion for prolonged-effect of CypA PROTACs

*Are the authors certain that the washout experiments worked? Once the ternary complex has formed and the target has been ubiquitinated, degradation of the target protein is usually fairly rapid. Figure 5's data showing that CypA degradation has not reached the maximum at 48 hours treatment and that the extent of CypA depletion is even greater following another 72 hours (120 hours total) is a puzzling result, in particular that the extent of depletion increases for 72h after the washout (Figure 5 condition 2). As noted above, the prolonged treatment times raise concerns that other mechanisms affecting CypA abundance are involved but are not being acknowledged. I also **don't think that the authors' explanation that additional CypA is being degraded by the PROTAC mechanism even when the PROTAC was washed out makes sense.** The prolonged pharmacological effect of PROTACs is generally due to the requirement for new protein synthesis to restore the function, which contrasts with the rapid regain of function upon dissociation of inhibitors. In Figure 5, it seems more likely that the washout was not effective. Therefore, I don't think that the claim that PROTAC-driven loss of CypA is long-lasting (lines 207-208) is solid **without better evidence that their washout worked.***

We appreciate the reviewer's feedback regarding the "washout" experiment, and we recognize the potential for confusion in its interpretation. Following this review, we have decided to remove the "washout" experiment from the manuscript, as it was intended to illustrate a different concept. In this context, "washout" referred to the replacement of the medium without the addition of the compound. However, this did not necessarily mean the compound had fully dissociated. This experiment was meant to illustrate CypA levels in T-cells upon continued treatment or non-continuous treatment. We agree that this experiment could be misleading and instead, we repeated the HIV replication experiment and monitor replication as well as CypA levels upon continuous and non-continuous treatments (new Fig. 6). We hope that this new figure presents more clearly the design of the HIV replication experiment.

4. Selectivity of CG167 and RJS308

a. Global selectivity

The authors address selectivity by comparing degradation of CypA and CypB; however, this analysis is inadequate to support the statement that CG167 and RJS308 are selective for CypA. It is well-established that PROTACs can induce degradation of neosubstrates that may not be predicted. As a consequence, the targeted protein **degradation field commonly utilizes** (and requires) **proteomic profiling experiments**, ideally performed with the PROTAC as well as with a negative control compound whose E3 ligand has been modified to prevent interaction with the E3 ligase. This analysis is of particular concern given the number of related cyclophilin proteins in the host cell. As recognized by the authors, the affinity of the interaction between the PROTAC and the target protein is not a predictor of the efficacy of target degradation. If the authors do not have in-house access to proteomic analysis, this can be accessed through collaboration and community efforts to define the degradable proteome.

We acknowledge the importance of proteomic analysis in this field and the need to ensure the specificity of PROTACs. In response to this comment, we conducted a proteomic profiling experiment to assess the impact of CG167 and RJS308 on the cellular proteome.

Our proteomic analysis reveals that CypA is the most significantly degraded protein by both PROTACs, with CypH and CypE degraded although less than CypA, and minimal effects on most other isoforms compared to the DMSO control (new Fig. 5b and c). Furthermore, the parental inhibitor TWH106 did not significantly affect the levels of any proteins (new Fig. 5a), with only a minor reduction in CypB levels, which is consistent with observations in our time-course and dose-response experiments (new Fig. 5e, i).

Considering the clean profile of the PROTACs, without significant depletion of neosubstrates, we have decided not to perform proteomic experiments with the PROTAC epimer for practical reasons. We believe that additional proteomic analysis of the epimer would not substantially contribute to the already compelling evidence of CypA selectivity.

b. Selectivity for CypA versus CypB

Although TWH106 was designed to be selective for CypA over CypB, SPR experiments with recombinant proteins indicate that this is not the case. While the authors mention that the lack of difference in the SPR experiments may be due to the use of **a commercially available, truncated form of CypB, it's unclear whether this is the case**. This is an important issue because the selectivity of the PROTACs for CypA over CypB may be due to a difference in affinity for these two targets and/or may be due to differences in ternary complex formation. **Is the superiority of the PROTAC over the parental inhibitor due to the rational change that was made to the ligand or is it due to differences in ternary complex formation and hence due to the PROTAC mechanism?** The authors should try to address this experimentally – **perhaps, CETSA experiments** monitoring engagement of CypA versus CypB in cellulo would be informative? They **should at least discuss these issues in the context of the PROTACs' selectivity for CypA versus CypB** when compared to the lack of selectivity for TWH106 and JW4-10. The explanation provided in **lines 192-196 is inadequate** given what is known about determinants of PROTAC selectivity.

We acknowledge that the previous explanation provided in lines 192-196 was not sufficiently detailed in the context of PROTAC selectivity. We have now expanded the discussion to consider various potential origins of CypA vs. CypB selectivity (lines 381-398). Furthermore, we have performed additional steps towards the elucidation of the CypB vs CypA selectivity mechanism:

1. Ternary complex formation by CypB: We performed an SPR experiment with the PROTAC RJS308 to assess the ternary complex formation of truncated CypB, indicating that CypB forms a sub-nanomolar, highly cooperative ternary complex with VCB. This data suggests that poor ternary complex formation between the cyclophilin domain of CypB, the PROTACs, and VCB is unlikely to explain the CypA over CypB selectivity (new Fig. 5d).

2. JW4-10 degrades CypB by a PROTAC-independent mechanism: New data also shows that the CypB level reduction observed with JW4-10 is likely a PROTAC-independent mechanism as MLN4924 and VH298 cannot rescue the degradation of CypA by JW4-10 (see reviewer #2 point 3b). As VHL is not involved, the reason behind CypA over CypB selectivity for VHL-mediated degradation seems independent of the warhead design.

Following the surprising selectivity of our molecules for CypA over most Cyp isoforms (new Fig. 5), we generated a bioinformatics figure (new supplementary Fig. 8) to illustrate and discuss various reasons for the broadly observed selectivity. In addition to direct affinity of the warhead to different Cyps, these include the influence of additional C- and N-termini flanking the Cyp domains preventing efficient ternary complex formation, subcellular localisation and protein accessibility for degradation or differential sequence at protein surfaces. We suggest that the interaction of these factors could impact compound diffusion times, PROTAC binding, ternary complex formation, and ubiquitination.

The observed VHL independent reduction of CypB levels at longer timepoints with TWH106 (most) and the PROTACs (to a lesser extent), indicate that binding of the compounds to CypB is likely occurring (line 398).

We acknowledge that fully understanding the mechanism of selectivity is a complex issue, and while CETSA could illuminate some of these hypotheses, a lot of these factors are challenging to prove experimentally. An in-depth study of these possibilities is beyond the scope of our paper, which primarily focuses on the effect of CypA depletion on antiviral replication. We believe that the added supplementary Fig. 8 and discussion in lines 381-398 provides a list of plausible explanations for the selectivity observed, while acknowledging the complexity of the issue.

5. Data in Figure 4i and 4j are not discussed in the text.

a. *What are the conclusions from these figures?*

Thanks for this observation, we have moved these to Supplementary Fig. 11h and updated the manuscript to discuss them (lines 276-277). The conclusions drawn from these figures concern the effectiveness of CypA/CypB degradation in primary resting and activated T cells. These findings underscore the utility of CG167 and RJS308 in various cell types and thus various antiviral models.

b. *Are the graphs in 4j derived from the blots in 4i or are the data in 4i from one donor and the data from 4j from another donor?*

The data in the graph in Figure 4j (new supplementary Fig. 11h) was indeed derived from the blots presented in Figure 4i (new supplementary Fig. 11g). We have updated the legend for new Fig. 5 and new supplementary Fig. 11 to explicitly mention this relationship, ensuring clarity for readers.

c. *Why do CG167 and RJS308 cause an increase in CypB in Figure 4j? Potential mechanisms should be discussed. Also why isn't the increase in CypB apparent in the 4j plots apparent in Figure 4i? I think that the plots combine data in Figure 4i with data from Supplementary Figure 7, but this should be explained in the Figure legend to make it clearer for readers.*

The increase in CypB levels observed with CG167 and RJS308 in Figure 4j (now new supplementary Fig. 11h), particularly in resting T cells, is likely due to an indirect, complex mechanism. This phenomenon appears to be highly variable, as indicated by the error bars.

We would like to emphasize that the main focus of our paper is the characterization of tools for studying the effect of CypA degradation on HIV-1 replication. As such, we have refrained from delving into an in-depth characterization of the mechanisms behind CypB level increases, which can be influenced by off-target effects in various cell lines. We appreciate the reviewer's feedback, which has contributed to our decision of moving these figures to the supplementary material. The CypA levels are quantified in-depth during the HIV replication experiment which made the inclusion of T-cell data in new Fig. 5 redundant.

Regarding the combination of data, you are correct that the plots in combine data from new Supplementary Fig. 11g (previously Fig. 4i and supplementary Fig._7). We have clarified the legends accordingly.

Minor:

1. a. In Figure 3, a control is needed showing the effect of MLN4924 along on CypA is needed. MLN4924 and proteasome inhibitors can lead to increased steady-state abundance of proteins. In this case, the neddylation inhibitor does not necessarily “rescue” the target protein from the PROTAC mechanism.

We acknowledge the importance of assessing the effect of MLN4924 on CypA independently and thus evaluated its sole influence on CypA levels. Our findings, presented in the new Fig. 4d and supplementary Fig. 6d, demonstrate that MLN4924 treatment does not alter the abundance of CypA. Moreover, to further support our investigation into PROTAC-mediated rescue mechanisms, we extended our experimentation to include VH298 (new Fig. 4e and new supplementary Fig. 6e).

Reviewer #3 (Remarks to the Author):

*This article by Gathmann et al. presents PROTACs aimed at Cyclophilin A (Cyp A), a protein cofactor exploited by viruses during infection. In this paper, the authors describe a PROTAC based on a new peptide macrocycle derived from Sanglifehrin A. This paper builds upon some of the authors previous work targeting CypA, this new heterobifunctional molecule is more specific for CypA over CypB and is another good example of the use of PROTAC as an antiviral treatment, and as the recent Covid19 pandemic has highlighted, new small molecules are needed. The authors demonstrated that at least one of their molecules acts through a PROTAC mechanism and was able to inhibit HIV-1 replication in human T cells. **The manuscript is well written, and the figures are particularly clear and readable. This manuscript should appeal to the readers of Nature Communications and should be published after some revisions and clarifications from the authors.***

1) Line 100: The authors mention SAR studies to develop the inhibitor TWH106. Was the SAR fully based on *in silico* docking or was **it based on SPR binding** as done with the PROTACs and the inhibitor as described later? It may be interesting to **mention it in the main text**.

We agree that mentioning the methodology used in the SAR study for the development of TWH106 would add valuable context to our findings. Initially, our SAR exploration included *in silico* docking studies to evaluate potential binding affinity. Subsequently, the designed compounds underwent validation through SPR binding assays to Cyps, with TWH106 one of the molecules showing high affinity and isoform selectivity to some extent. These steps in the inhibitor development process were not explicitly detailed in the initial manuscript due to considerations of scope, but we have now incorporated a concise mention of this approach in the revised manuscript (lines 114-115).

2) Line 118-119: The authors focus their PROTAC synthesis on very specific linkers, namely linkers of 10 or 12 atoms with a triazole stemming from the cycloaddition of the two halves of the molecule. In heterobifunctional molecules, linker length and composition play a major role in degradation efficiency. In their previous report (Colpitts et al. eLife 2002) of a CypA PROTAC using cyclosporin A as a warhead, **the linker used is of only 3 atoms**. How did the authors choose this specific linker length and were several other linkers (length or composition) tested? It may be worth mentioning the author's efforts to develop the most potent PROTAC possible. On the same topic, were other E3 ligases considered during the PROTAC development, namely using thalidomide derivatives or other attachment points on the VHL ligand?

We appreciate the interest in considerations in the development of PROTACs targeting CypA. The selection of specific linker lengths for our PROTAC molecules, namely 10 or 12 atoms with

a triazole linkage, was primarily guided by practical considerations during a focused, short-term effort amid the desire to rapidly develop antiviral strategies during the pandemic. Notably, our design strategy involved utilizing commercially available linkers with orthogonal chemistry to our molecules, leading to the adoption of the cycloaddition approach rather than employing cross-metathesis linkers as used in JW4-10. The decision to opt for linker lengths of 10 or 12 atoms was influenced by our prior experience, particularly with JW4-10, our most promising CsA-based PROTAC candidate. In JW4-10, the linker length can be counted as 10 between the core macrocycle and the VHL ligand, considering the protruding double bond of CsA from the active site (see illustration in supplementary Fig. 8b).

JW4-10, CG167, RJS308 structures with linker region in red

We added a statement in line 138-140 explaining the inspiration from JW4-10. For comprehensive clarity, the complete chemical structure of JW4-10 is now also available in supplementary Fig. 10a.

Regarding the consideration of alternative linkers or E3 ligases during PROTAC development, it's pertinent to note that our past, albeit unpublished, investigations revealed that cereblon-directed CsA PROTACs did not exhibit the desired efficacy as CypA degraders. This influenced our decision to use VHL ligand 2 for our current study. We have incorporated this significant information into the revised manuscript (lines 134-135) to further elucidate our molecular design process.

3) Figure 3 and CypA degradation by a PROTAC mechanism:

a. The authors demonstrate that their molecules act by PROTAC mechanism by using a NEDD8ylation inhibitor and by competition with free inhibitor. In these degradation blots, each data point has been tested as a single technical replicate and as a biological duplicate. In some cases, Figure 3a and 3b for CG167 and Figure 3b for RJS308, the degradation levels vary from 50% to close to 100%. A third biological replicate would be beneficial to conclude on the degradation potential of these molecules, since the two data points are so different.

b. It looks like in the experiment 2, the degradation levels observed are lower across all conditions tested, could this experiment be an outlier?

Following this suggestion, we performed the third replicates for all experiments in this figure and calculated adjusted relative density from the western blots. We have added this data to the manuscript (new Fig. 4a-e and supplementary Fig. 6). Considering this new data, and also other experiments where we have quantified CypA degradation at 48 hr timepoint following PROTAC treatment, we concluded that experiment 2 (from the original manuscript) was an outlier, as suggested by the reviewer. We have therefore removed this outlier experiment from the new manuscript.

c. The authors mention that a ternary complex between CG167, CypA and VCB could not be

observed. Moreover, in supplementary Figure 5b: Was the competition FP experiment not run with CG167 or was no competition observed? *The lack of binding of CG167 to VHL would suggest that this compound may be able to degrade CypA through a non-PROTAC mechanism. Has the ability of this PROTAC to bind VHL been verified by the authors?*

We appreciate the inquiry regarding the ability of CG167 to bind VHL and thus form a ternary complex between CypA and VCB and agree it needed clarification. In response to this query, we revisited and updated the data in supplementary Fig. 7, where the competition Fluorescence Polarization (FP) assessing direct binding between CG167 and VHL is shown. The binary complex between CG167 and VHL is observable by FP, albeit only at abnormally high concentrations (>2.5 μM), while SPR could not show substantial binding at all. This observation strongly suggests that the nature and positioning of the linker profoundly influenced the binding affinity of CG167 to VHL. The compound structure was extensively verified by NMR 2D studies, and the VHL epimer starting material was not in our possession, rendering wrong stereochemistry of the VHL ligand unlikely.

Nevertheless, our study indicates that CG167, although displaying a slower degradation profile, still exhibits CypA degradation through the proteasome and is contingent upon VHL ligase activity (new Fig. 4 and see also reviewer #2 point 3b). These findings collectively support the notion that CG167 functions as a 'slow' PROTAC rather than operating through a non-PROTAC mechanism, where only a very small portion of CG167 engages VHL but produces an effect at longer time points due to the catalytic nature of PROTACs.

However, acknowledging this and reviewer #2 point 3b, as well as the complexity of assessing PROTAC mechanisms comprehensively, we agree that the possibility of alternative mechanisms cannot be categorically excluded. We think the additional experiments showing rescue by VHL inhibitor VH298 support a VHL dependent mechanism, but we acknowledge that the lack of *in vitro* engagement is surprising and are working on elucidating this, as stated in manuscript line 378.

4) CypA degradation by a PROTAC mechanism and Degradation of cyclophilins A and B (Figure 3 and Figure 4):

- a. *The authors do not show any clear dose response or time course of their PROTAC activity. How were the treatment conditions selected?*
- b. *In their previous paper (Colpitts et al. eLife 2002), the authors showed that their previous PROTAC was able to degrade CypA, as low as 30 nM and that 48h treatment was only required to reach 100% degradation. Have the authors compared the two new PROTACs to their previous molecule in a time course or in a wider dose response?*
- c. *The SPR experiment shows about a 100 nM affinity of the PROTACs for their target, is 1 μM treatment needed or is this due to poor cell permeability of the molecules?*

In response to this comment and to help rationalise our choice of treatment parameters, we have performed a comprehensive characterisation of the time and concentration behaviour of our Cyp-PROTACs in the main manuscript (new Fig. 3), which was initially missing, as rightly pointed out. Reviewer #2 had similar comments in points 1b and 2, we paraphrase our rationale here again:

1. Treatment time course: Our study employs Cyp-PROTACs as tools to investigate the influence of complete CypA removal on viral infection, requiring maximal CypA degradation. The slow action of CG167 (new Fig. 3g and see reviewer #2 point 3b) prompted us to use 48h minimum treatment time. Importantly, our data shows that the parental inhibitor TWH106 has no significant effect on CypA levels over a 48h treatment at 5 μM .

2. Proteomics experiments (new Fig. 5a-c): We conducted proteomics experiments after a 72-hour treatment to assess off-target degradation for all compounds tested. The results indicated minimal off-target effects, further supporting our choice of a 48-hour treatment duration.

3. Dose response experiments: Dose-response experiments were performed at both 24h and 48h treatment times. These experiments revealed a DC_{max} of approximately 1 μ M for CG167 and 5 μ M for RJS308. In contrast, the parental inhibitor TWH106 exhibited a mild CypA loss only at 10 μ M and after 24 hours (new Fig. 3n). Given the plateau observed in CypA degradation at 5 μ M for both PROTACs and the absence of off-mechanism CypA degradation for TWH106 at this concentration, we selected 5 μ M for subsequent experiments.

In summary, our choice of a 48h treatment time and 5 μ M concentration for Cyp-PROTACs is well-justified based on the specific goals of our study, the lack of off-target effects observed, and the high abundance of CypA as a cellular protein. These considerations collectively support the rationale for our experimental design, and enable us to assess both PROTACs simultaneously, thus reducing practical expenses.

5) *Degradation of cyclophilins A and B (Figure 3 and Figure 4):*

a. *When comparing the degradation selectivity of their PROTACs, the authors show that JW4-10, a cyclosporin A based PROTAC can degrade CypB. As mentioned earlier, in the biological replicate of this experiment, degradation of CypB was observed to a much lower level. A third data point would be welcome to validate the extent of CypB degradation.*

These experiments were repeated completely as part of a new figure characterising JW4-10 mediated degradation of CypA and CypB in Jurkat cells and their rescue by MLN4924 and VH298 (Supplementary Fig. 10). In those experiments, CypB degradation was consistent across replicates, clarifying the extent of CypB degradation.

b. The authors mention the weaker degradation potential of their PROTACs in U87 cells, but they never provide any explanation as to why this is observed. Can the authors comment on this observation?

We appreciate the keen observation regarding the comparatively weaker degradation potential of our PROTACs in U87 cells and the inquiry into potential explanations for this observation. Upon investigating this discrepancy, we explored the possibility of lower VHL ligase levels contributing to the observed differences but our data (supplementary Fig. 12g) did not indicate reduced VHL ligase amounts as the underlying cause. We provide a statement in lines 341-342 about the VHL levels.

Regrettably, at present, we do not possess definitive insights into the precise reasons behind the observed variation in degradation potential across different cell types. It seems likely that the simplest explanation is differential cell permeability. We recognise the interest and significance of understanding these variations; however, we believe that elucidating these mechanistic details extends beyond the primary scope of this paper.

6) Figure 6: Why was the time of pre-incubation between U87/Jurkat cells and donor T-cell changed from respectively 48 hours to 72 hours?

The set-up of these experiments was initially designed to maximise CypA removal from the cells prior to viral infection experiments. However, following this comment and the timecourse data (new Fig. 3) we have decided to repeat the antiviral experiments matching the 48h of the previous experiment (new Fig. 6), as the additional degradation observed does not justify the change.

7) *Supplementary Figure 2: SPR experiment. In this SPR experiment, the theoretical maximum RU expected is close to 220 RU. Can the authors comment on why such a high density of protein was needed? Despite having a similar size, the two PROTAC show very different binding behaviors, namely RJS308 binds with a maximum RU of 130, while CG167 only binds with a 65 RU maximum. Do these PROTAC bind in different pockets, or do they bind a different subset of the protein population immobilized on the surface? What does this weaker RU entail for CG167?*

Upon reviewing the data, we realised an error where the second plot, supposedly representing CG167, was, in fact, a repetition of TWH106 on a different chip. We have taken prompt corrective measures and replaced this dataset with the correct one, which now accurately portrays the binding behaviour of CG167, showing a maximum RU comparable to that of RJS308. We have ensured the availability of all raw data, including the previous plot, in the associated data repository link associated with the paper for complete transparency and reference. Consequently, this rectification slightly alters the mean K_D values of CG167 and TWH106, which have been updated in Table 1. Importantly, this correction does not alter any of the conclusions drawn in the study. Thank you for this observation, which prompted us to rectify this error.

8) Supplemental information: The addition of the NMR spectrum for each small molecule synthesized on top of the peak report would be appreciated for full transparency. However, since the authors did a thorough analysis of their final compounds, the NMR spectra may not be required and there is no doubt that they synthesized the desired molecules.

To improve transparency in the medicinal chemistry field, we decided to take this step and include all NMR spectra of intermediates. Thank you for the suggestion.

October 2024 Response to Reviewers Letter

To the editor

We would like to thank again the reviewers for the assessment of the manuscript and their contributions. We are thankful to them for recognising the considerable efforts put into performing additional experiments to improve the manuscript. Our responses to the reviewer comments are provided below in red. We have also included a comparison of the re-submitted manuscript (August 2024) and this new version.

We have addressed the concern of 'supporting antiviral effect data for PROTAC vs inhibitor' by explaining more clearly the set-up of our antiviral experiments, which may have led to confusion initially. We tone down our conclusions by explaining that the PROTACs are better antivirals in specific contexts. The manuscript discussion has been updated and presents these molecules more clearly as proof of concept for the development of PROTAC antivirals and to study the role of CypA in viral replication rather than clinically relevant therapies.

Concerning previously raised points by the reviewers, we believe we have performed all requested additional experiments, including time-course data, dose data, proteomics data, addition of NMR data and SPR data for ternary complex formation. Where we have refrained from performing experiments (CETSA data for CypB engagement in cells in reviewer #2 comment 4b) we clearly explain why. We also included a more detailed answer specifically tailored to comment 4b and c of reviewer #3 at the end of this letter, explaining how the newly added experiments and the modified text address these points, as we realised that the original answer was a more general reply to comment 4. For clarity, we have also included the first rebuttal letter.

We believe these modifications clarify the conclusions of our manuscript and answer the newly raised concerns by the reviewer.

We look forward to your answer,

With kind regards

David Selwood

(on behalf of all the authors).

REVIEWER COMMENTS

Reviewer #2 (Remarks to the Author):

The authors have done a tremendous job adding in new experiments to demonstrate that CG167 and RJS308 act via a TPD mechanism. I think this is a great study; however, I don't think that all of the conclusions/claims are supported by the data. I recommend acceptance if the following are addressed:

Major issues:

1. Since cyclophilin H and E came up as significant targets of both PROTACs, the authors should provide data showing whether either or both contribute to the antiviral activity of CG167 and/or RJS308. CypE has come up as a restriction factor for influenza viruses and given its nuclear localization could affect HIV. Alternatively, authors should add text acknowledging that this is a possibility and that their reported experiments do not bear on this.

The reviewer's suggestion has been taken into account and we have included the following text in the discussion:

“In this study we have focused on CypA because it was most sensitive to PROTAC degradation in proteomic experiments (Fig. 5). However, these experiments also revealed degradation of CypE and CypH. While CypA has an established role as a viral cofactor, CypE and CypH have also been suggested to potentially be involved in HCV replication. We cannot rule out the possibility that degradation of these proteins contributes to the antiviral activity of our Cyp-PROTACs.”

2. The authors claim that the Cyp PROTACs have greater potency at lower concentrations compared to the Cyp inhibitor. I think this is true at a single concentration (1 μ M) and a single time point (day 4); however, the PROTACs have lower potency as measured by reduced viral RNA at 5 μ M on both days 4 and 8, and antiviral efficacy at the effective dose is also LOWER for the PROTACs. The antiviral activities are the same for the PROTACs and the inhibitor at the last time point in the Gag+ assay (Figure 6b-6e, 6n-6o) and the RT assay (Fig 6r-6s). The inhibitor also looks more potent and/or more efficacious in Supplementary Figures 12b-12c, 13a-13b, 14a-14c (some but not all plots). It's hard to interpret the data because only two concentrations are being reported, the abundance of CypA (or CypE and CypH) in these cells is not known, and the time points are long. A titration curve for the antiviral activity is really what's needed here along with immunoblot (or perhaps intracellular staining quantified by flow cytometry) data indicating how much CypA has been depleted. I realize it's hard to conduct these experiments, but then the conclusions need to be moderated to avoid over-interpretation of the data.

The reviewer has made the point that the PROTACs are not more effective inhibitors in every circumstance. This is true and we discuss this in the discussion section. We have now clarified this with the following text arguing that PROTACs work better when inhibitor is limiting and in primary cells:

“We find that Cyp-PROTACs have antiviral activity against HIV-1 replication in primary human CD4+ T cells (Fig. 6) and against HCV replicon in Huh7 cells (Supplementary Fig. 16). We find increased antiviral activity of PROTACs over parental Cyp inhibitor TWH106 in specific circumstances, most importantly when inhibiting HIV-1 replication in primary T cells with lower inhibitor concentration (1 μ M), or when the inhibition duration was reduced to pretreatment only. We hypothesise that at high concentration (5 μ M) parental TWH106 is more effective than the PROTACs because the PROTAC modifications affect cell permeability and

reduce target engagement. However at lower concentrations (1 μM), when inhibitor is limiting, a PROTAC catalytic degradation mechanism has improved antiviral activity, compared to active site competitive inhibition. Similarly, when cells are pretreated with inhibitors, HCV replicon replication is suppressed more effectively by parental inhibitor TWH106 at high dose (5 μM), while at lower dose (1 μM) the PROTACs are superior. PROTAC-mediated degradation may also gain antiviral activity through loss of target function unrelated to the active site. It is true that parental TWH106 is a more effective inhibitor than the PROTACs in cell lines models (Supplementary Fig. 12a-f). While we do not understand what differs between the cell lines and the primary T cells, we assume that the primary T-cell experiments represent the situation in vivo better than the cell lines. Notably, it has previously been shown that the ability of CypA to protect HIV-1 from restriction by TRIM5a is more pronounced in primary cells.”

As the reviewer points out, it is true in some experimental set ups, at particular inhibitor doses, the maximum inhibition of infection is achieved by the parental non-protac inhibitor. However, we see improved antiviral activity at lower concentrations, or at shorter inhibitor exposures and discuss this in the text above.

The reviewer highlights the importance of measuring CypA abundance. In fact, we have done this, see Fig. 6j-m and Fig. 6v-x, where CypA abundances are measured and clearly demonstrate PROTAC-mediated CypA degradation. Similarly, this data is included in Supp. Fig. 14.

The reviewer refers to Fig. 6n-o and Fig. 6r-s. This is the summarised data of 4 donors at 4 days post infection (i.e. NOT data from the last timepoint), and illustrates that at 1 μM , PROTACs are more antiviral than the inhibitor, whereas at 5 μM , the inhibitor is more antiviral than the PROTACs. Regarding inhibitory levels being the same at the last timepoint in Figure 6b-6e, we address this in our response to point 6 (see below).

3. Likewise for the HCV antiviral data: Authors need to discuss that the inhibitor has greater antiviral efficacy at its effective dose (10 μM) than the PROTACs. It would be better if they had a titration and could discuss the EC50 values of the PROTACs versus the inhibitor.

This is addressed above in our response to point 2.

4. Time points: Shouldn't Cyp inhibition happen upon binding and so antiviral activity due to loss of CypA function should happen with the same or faster kinetics than antiviral activity due to targeted degradation of CypA induced by the PROTAC? Why in all of the time courses does the inhibitor's antiviral activity lag that of the PROTACs? This raises the concern that the antiviral activity is not due to loss of CypA activity. Does CypA have a scaffolding function that is not affected by the inhibitor but is affected by the PROTAC?

We agree that effects beyond the active site inhibition may also play a role in antiviral activity. We acknowledge this in the text:

“PROTAC mediated degradation may also gain antiviral activity through loss of target function unrelated to the active site.”

We also suggest that these molecules will be good tools to study this specifically: *“the cleaner profile of PROTACs CG167 and RJS308 compared to other inhibitors could simplify the investigation of the roles of different cyclophilins in viral infections and beyond. For example, our TWH106 series may be useful to distinguish the effect of CypA inhibition at the PPlase active site versus protein removal”*

We would like to clarify that kinetic effects of the inhibitors are not visible in our timecourse experiments (Fig. 6), as the cells were pretreated with inhibitors prior to infection. CypA was therefore depleted by the PROTACs (as visible in Fig.6j-m) before the cells were infected. Nevertheless, for the 5 μ M treatment condition (Fig.6c,g), we observe that TWH106 is in fact more antiviral than the PROTACs (i.e. not ‘lagging’ behind as suggested by the reviewer).

5. Description of the HIV-1 experiments is confusing. What was the viral challenge? Multiplicity of infection should be reported. “....cell swere mixed and incubated with 2000 mU reverse transcriptase of HIV-1 NL4.3” doesn't make sense. Did you transfect in viral RNA or did you infect?

Using RT levels as a measure of HIV input dose are typical, although we agree they do not necessarily help us understand the starting MOI. Measuring input MOI at the beginning of a spreading infection is typically hard to do because only a few cells get infected and then the infection spreads through the T-cell culture. Considering the flow cytometry data measuring Gag expression one can see that we don't see any infection until day 4 by which time infection is tens of percent (15-35%). This is not input MOI, which is much lower. We have clarified this in the methods with the following text:

“To infect activated primary CD4+ T cells with HIV-1, cells were mixed and incubated with HIV-1 NL4.3 virus (2000 mU RT/10⁶ cells) for 4 hrs at 37 °C. This is effectively a low MOI, and percentage infection (% Gag+ cells) varied between 15 % and 35 % at 4 days post infection (dpi), depending on the donor.”

6. Lines 73-79: It's not clear that PROTACs can address the liabilities that prevented advancement of CypA inhibitors against HCV and HIV-1. The Cyp inhibition had adequate potency, the limiting issue has been that the viruses are not as dependent on CypA activity. Also, as one gets into the data presented, TWH106 and the two PROTACs have equivalent antiviral activity at later time points in Figure 6 and antiviral efficacy (max inhibition of virus) is higher for the inhibitor. This language needs to be moderated.

We do not propose these PROTACs as clinical therapies for HIV and HCV. Apart from anything else, these viruses have fantastic inhibitors in the clinic and CypA PROTACs would be unlikely to improve on these. Here, we use Cyp PROTACs as an experimental system to test whether PROTACs can make better antivirals,

supporting this notion at low dose inhibitor, as explained above. We have made this clearer by including the following text in the discussion:

“These data generally support the development of PROTACs as antivirals, with PROTACs giving particular advantage at lower doses or when exposure to inhibitor is reduced.”

The idea that later time points illustrate similar antiviral activity is only true for the measurement of Gag by flow cytometry (e.g. Fig. 6b-c). This is heavily influenced by cell death, which strongly suppresses Gag positivity by the end of the experiment. For measurements of RT in the supernatant (e.g. Fig. 6f-g), which is representative of released viral particles, the inhibition by the parental TWH106 does not catch up. We argue these data are consistent with our conclusions and we provide further explanation in the results as follows:

“Notably, at late time points (8 dpi), Gag expression measured by flow cytometry was suppressed as a result of cell death, leading to similarly low % Gag+ measurements. However, for measurements of infectious virus in the supernatant (RT), differential antiviral activity was observed at these later timepoints.”

Minor questions:

7. Sentence "Antiviral PROTAC development has also gained interest for its potential to reduce sensitive to resistance" should cite reference 72 as well since this is the one publication showing this can be possible.

This reference has now been added, as suggested by the reviewer.

8. Do modeling and MD simulations support the hypothesis that the exit vector is the issue for CG167?

The essential difference between PROTACs CG167 and RJS308 is in the position of the linker relative to CypA, as the VHL side is the same in both cases. The affinity to CypA is retained for both (Table 1) but we cannot detect binding to recombinant VHL protein at 10 μ M by SPR (the maximum tested) and some weak engagement occurs above 10 μ M by FP assay. Both PROTACs make approximately similar, into solvent, vectors, so it remains unclear what underlies their different activities. Perhaps molecular dynamics simulations and structural studies could help recognising more complex effects of slight changes in linker structure in the future, but we think this is beyond the scope of the current study. Notably, we would like to emphasise that engagement of VHL by CG167 must occur in cells as the degradation is prevented with VH298.

We discuss this in the following text:

“While VHL engagement in cells is demonstrated by the rescue of CG167-mediated CypA degradation with VH298 (VHLi Fig. 4b), its slower kinetics (Fig. 3a-g) are consistent with reduced VHL engagement and poor ternary complex formation measured in vitro (Fig. 4g). The molecular basis for this is still unclear, as RJS308 and CG167 have identical attachment positions to the VHL binding moiety and differ only slightly by

linker structure. Certainly, previous studies have shown that small changes in PROTAC linker can drastically affect E3 ligase affinity”

Some comments on figures and legends – the data are very nice, but hard to take in. These suggestions are meant to make it easier for your readers:

9. Figure 4a-e: Indicate what time post-infection the samples were collected on the graph or provide schematics to help readers understand the different treatment conditions. It's written in the legend, but difficult to follow.

Thank you for your suggestions for making our figures clearer. Regarding Fig 4a-e: there is no infection in these rescue experiments, thus we cannot include time points post infection as suggested. However, we have edited the caption to make the different treatment conditions clearer:

“a-e Immunoblots detecting CypA, or b-actin as loading control, in Jurkat cells. Cells were pretreated with (a) 1 μ M NEDD8-activating enzyme inhibitor MLN4924 for 6 hrs or (b) 50 μ M VHL inhibitor VH298 (VHLi) for 2 hrs, followed by 5 μ M PROTAC CG167 or RJS308 for 48 hrs, or cells were treated with (c) 10 μ M TWH106 and 1 μ M PROTAC CG167 or RJS308 for 48 hrs, or cells were treated with (d) MLN4924 as in (a), or with (e) VHLi as in (b).”

Figure 6:

10. It was difficult to discern the grey, green, and blue colors on my screen given the size of the plots. Suggest that the authors try to make thicker lines, use colors with greater contrast, and/or add legends for the symbols to the graph. Legend should indicate multiplicity of infection or some other metric for the size of the viral challenge in this experiment.

To make it easier for readers to distinguish the different treatments in Fig. 6, we have changed the colours of symbols and lines to be more contrasting (TWH106 now in orange) and colour-blind friendly. The legend is included at the top of the figure. We have changed other graphs in the paper correspondingly for consistency. Increasing the thickness of the lines made it harder to distinguish between overlapping lines, so we did not make this change. We have added a metric for viral challenge (2000 mU RT/10⁶ cells) to Fig.6 caption as suggested.

We also include additional clarification regarding points previously made by reviewers:

Reviewer 2, comment 4:

CypA degradation by a PROTAC mechanism and Degradation of cyclophilins A and B (Figure 3 and Figure 4):

b. In their previous paper (Colpitts et al. eLife 2002), the authors showed that their previous PROTAC was able to degrade CypA, as low as 30 nM and that 48h treatment was only required to reach 100% degradation. Have the authors compared the two new PROTACs to their previous molecule in a time course or in a wider dose response?

To address this, we included a 6-day timecourse experiment and dose response experiments (Figure 3) for PROTACs CG167 and RJS308 in the manuscript. We

have not directly compared these PROTACs to our previously reported PROTAC JW4-10 (CsA-Prtc1), however, comparing with our previous paper (Colpitts et al. eLife 2002), we find that higher concentrations of PROTACs CG167 and RJS308 are required for effective CypA degradation, compared to JW4-10. We believe the newly added timecourse and dose data supports our concentration and time choice as justified in the response to comment 4a.

c. The SPR experiment shows about a 100 nM affinity of the PROTACs for their target, is 1 uM treatment needed or is this due to poor cell permeability of the molecules?

In line with the newly added timecourse and dose data discussed in response to comment 4a and b, we have chosen concentrations to be effective in cell models. The requirement for higher concentrations in cells may indeed be due to poor cell permeability of the molecules, as rightly pointed out by the reviewer. Alternatively, the high abundance of CypA in cells may explain such discrepancies between in vitro data and cellular data. We briefly point out to this possibility in the text:

“High PROTAC concentrations sometimes reduce activity through titration of the ternary complex, the so-called hook effect, but we did not observe this. This is perhaps linked to the high abundance of CypA in cells”.